# Leveraging Shared Prototypes for a Multimodal Pulse Motion Foundation Model

## Abstract

Modeling multi-modal time-series data is critical for capturing system-level dynamics, particularly in biosignals where modalities such as ECG, PPG, EDA, and accelerometry provide complementary perspectives on interconnected physiological processes. While recent self-supervised learning (SSL) advances have improved unimodal representation learning, existing multi-modal approaches often rely on CLIP-style contrastive objectives that overfit to easily aligned features and misclassify valid cross-modal relationships as negatives, resulting in fragmented and non-generalizable embeddings. To overcome these limitations, we propose ProtoMM, a novel SSL framework that introduces a shared prototype dictionary to anchor heterogeneous modalities in a common embedding space. By clustering representations around shared prototypes rather than explicit negative sampling, our method captures complementary information across modalities and provides a coherent "common language" for physiological signals. In this work, we focus on developing a Pulse Motion foundation model with ProtoMM and demonstrate that our approach outperforms contrastive-only and prior multimodal SSL methods, achieving state-of-the-art performance while offering improved interpretability of learned features.

## 1 Introduction

Digital biomarkers (for stress, physical activity, sleep, etc.) obtained from wearable sensors, such as smart watches and smartphones, provide unprecedented opportunities to give individuals novel insights into their states of health and wellness throughout their daily life, along with new tools for managing their health-related behaviors (Rehg et al., 2017). In order to realize this potential, however, it is critical to develop effective models for *multi-modal* time series biosignal data, so that complementary sensing modalities can be leveraged to overcome the ambiguities and noise that are inherent in wearable signals collected in the field environment.

Recently, there has been substantial progress in developing unimodal Foundation Models (FMs) which are pre-trained using large datasets on modalities such as accelerometry (Xu et al., 2024b; Yuan et al., 2024), ECG (Abbaspourazad et al., 2023; McKeen et al., 2024), and PPG (Saha et al., 2025; Pillai et al., 2024). These models have demonstrated effective generalization to downstream tasks and have established new benchmarks for performance. Building on these successes, recent works have focused on the challenge of how to align multiple signal modalities in pretraining multi-modal FMs, often using CLIP-style contrast objectives that pull temporally aligned signals together (Thapa et al., 2024; Deldari et al., 2022; 2024; Zhang et al., 2025). A key challenge is to ensure that the resulting multimodal embedding captures both *between-modality* information (i.e., features shared across modalities, such as the features of the cardiac cycle that are present in all cardiovascular signals such as ECG, PPG, and ICG) and *within-modality* information (i.e., features that are unique to a single modality, such as the signatures of kinematic motion that are present in accelerometry). When modalities are highly complementary, such as PPG and accelerometry, there is a danger that the alignment process could emphasize between-modality features at the expense of within-modality features. This is because cardiovascular activity (as captured in the PPG signal) is only indirectly connected to kinematic movement (as captured via accelerometry), e.g. through the increase in cardiac activity which accompanies strenuous exercise. Emphasizing signal alignment could inadvertently discard information which is unique to each modality and critical for downstream tasks such as stress detection.

This paper introduces **ProtoMM**, a novel self-supervised alignment strategy for pre-training a pulse motion foundation models with PPG and accelerometry signal modalities. The goal of ProtoMM is to test the hypothesis that alignment of complementary signal modalities can be facilitated via a prototype-based approach, in which biosignals are discretized into prototype vectors as part of the embedding process. We hypothesize that these prototypes will be effective in encoding within-modality features and preserving them during the embedding process. ProtoMM achieves this goal by first creating multiple augmented views from each modality. Embeddings from all views, derived using different augmentations of the same modality or from different modalities entirely, are then projected onto a shared set of prototype vectors. The model is trained using a Multimodal Prototype Prediction loss, where the prototypes probabilities of one view must predict the prototype assignment of another. By enforcing this consistency across all pairs of views, the prototypes become discrete, learnable anchors for the shared latent space for both within-modality and between-modality information. We develop and test ProtoMM for the task of joint multi-modal modeling of PPG and accelerometry data. This is beneficial because PPG can be used to detect the physiological stress response through cardiovascular changes (Jahanjoo et al., 2024), while integrating accelerometer data is essential to disambiguate between responses due to physical activity versus responses caused by psychological stress (Sevil et al., 2020; Sun et al., 2012).

We validate the potential of ProtoMM via thorough experimental evaluation that first demonstrates how ProtoMM captures within and between modality information. Next, we show how explicit modeling of latent states via discrete prototypes is particularly useful in our multimodal setting. With this, ProtoMM achieves superior performance against state-of-the-art multimodal self-supervised learning methodologies, and we can qualitatively validate that our prototypes capture morphological similarities with higher-level semantic information. We will release our model weights and a codebase with the full training methodology, architecture, and reproducible evaluation code, upon acceptance. The main contributions of this work are:

1. We introduce **ProtoMM**, a novel multimodal self-supervised framework for pulse motion foundation model, that resolves a key limitation of existing alignment methods. By using a shared set of prototypes and a swapped prediction objective, our model is designed to capture both within-modality (unique) and between-modality (shared) information.

2. We conduct extensive experiments by evaluating its transferability on three downstream datasets across six distinct tasks. We obtain superior performance, showing that ProtoMM consistently outperforms leading multimodal and unimodal baselines.

3. We demonstrate that the explicit nature of our prototype-based learning leads to improved interpretability. Through qualitative analysis, we show that individual prototypes can learn to represent specific, semantically meaningful physiological and behavioral states.

## 2 RELATED WORK

In recent years, learning useful representations from unlabeled sensor data has become the predominant paradigm, leveraging the ease of wearable sensors to record large quantities of data in naturalistic conditions (Bycroft et al., 2018). Popular approaches include future prediction (Narayanswamy et al., 2024; Haresamudram et al., 2021), contrasting between randomly augmented segments (Tang et al., 2020; Haresamudram et al., 2022), probabilistic transformation prediction (Saeed et al., 2019; Yuan et al., 2024), and reconstruction of randomly masked data (Haresamudram et al., 2020; Narayanswamy et al., 2024; Xu et al., 2025; Miao et al., 2024).

**Contrastive Representation Learning for Time-Series Data:** Prior work has demonstrated that contrasting randomly transformed windows of sensor data is highly effective, e.g., SimCLR (Tang et al., 2020). Similarly, motif-based positive pair generation for contrastive training has shown great promise (Xu et al., 2024a;b). Using data from a single modality, these methods essentially learn *within-modality information*.

Alternatively, approaches like ColloSSL (Jain et al., 2022), CroSSL (Deldari et al., 2024), and COCOA (Deldari et al., 2022) mine positives and negatives *across sensors and modalities*. They evaluate using diverse modalities, including accelerometers, gyroscopes, ECG, EMG, and EDA. As such, a critical drawback is the non-trivial nature of mining the pairs. More recently, aligning sensor data with natural language descriptions has emerged as an effective option, essentially adopt-

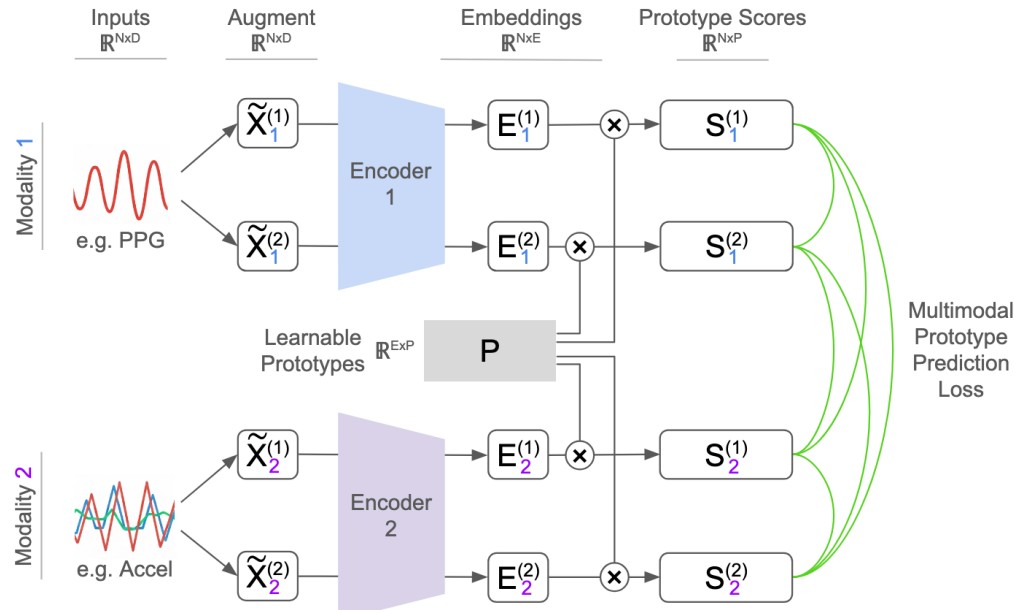

Figure 1: ProtoMM processes augmented segments from multiple modalities (i.e. PPG and Accelerometry) through dedicated encoders to produce the embeddings. The embeddings are then projected onto a shared set of prototype vectors, and the model is trained with a Multimodal Prototype Prediction Loss ($\mathcal{L}_{\text{MPP}}$) that learns to capture both within- and between modality information without relying on negative sampling.

ing the CLIP framework (Radford et al., 2021) for time-series data. Methods such as IMU2CLIP (Moon et al., 2022), Ts2Act (Xia et al., 2024), FOCAL (Liu et al., 2023), and SensorLM (Zhang et al., 2025) have demonstrated the capabilities of such modeling. Further, SLIP (Mu et al., 2022), which is a combination of the SimCLR and CLIP objectives was evaluated in Haresamudram et al. (2025), showcasing improvements. As such, a majority of these methods model only the common *between-modality information*. Our approach–ProtoMM–models both within- and between-modality information. Further, it sidesteps the challenges associated with mining positive/negative pairs by enforcing consistency between cluster assignments of augmented inputs.

**Prototype-Based Representation Learning:** Instead of instance-based discrimination used in approaches like SimCLR, SwAV (Caron et al., 2020) jointly clusters the data using prototype vectors, and enforces consistency between soft cluster assignments produced by different augmented inputs. VQ-VAE (Van Den Oord et al., 2017) also employs such vectors, and combines an autoencoder with vector quantization in order to perform online clustering with hard assignment. This setup has been extended to pose data as well (Zhang et al., 2023; Wang et al., 2024). Instead of the autoencoder, VQ-Wav2vec (Baevski et al., 2019; 2020) and VQ-CPC (Haresamudram et al., 2024) use CPC (Oord et al., 2018) as the base. As such, vector quantization based methods perform hard assignments of the prototype vectors. Our work builds on SwAV, which performs soft cluster assignments, leading to richer expressivity as there can be semantic overlap within the prototype vectors.

## 3 PROTOMM: METHODOLOGY AND DESIGN

We introduce ProtoMM, a multimodal self-supervised learning framework for pre-training pulse motion foundation model, which learns semantically meaningful representations by aligning multimodal time-series data to a shared set of prototypes. Our approach generalizes the swapped assignment prediction mechanism, originally proposed in the unimodal, two-view setting by SwAV (Caron et al., 2020), to a more general setting involving an arbitrary number of modalities as well as views per modality. By enforcing both within and between-modal consistency, the model learns features that are robust to augmentations while being coherent across different sensor modalities.

## 3.1 PROBLEM SETUP AND NOTATION

We define a multimodal time-series as $\mathbf{X}_t = \{\mathbf{X}_{t,1}, \mathbf{X}_{t,2}, \ldots, \mathbf{X}_{t,M}\}$, where $M$ is the number of sensor modalities and $\mathbf{X}_{t,m} \in \mathbb{R}^{T_m \times C_m}$ represents the data for the $m$-th modality. The subscript $t$ indicates that all modal data are temporally aligned to the same window; we omit it for brevity when the context is clear. The dimensions $T_m$ and $C_m$ allow for varying sampling frequencies and channel sizes across modalities.

A data augmentation module, $\mathcal{A}(\cdot)$, is used to generate $A$ distinct views for each modality, creating an augmented set $\{\tilde{\mathbf{X}}_m^{(a)}\}$ for all modalities $m \in \{1, \ldots, M\}$ and views $a \in \{1, \ldots, A\}$. Each augmented view $\tilde{\mathbf{X}}_m^{(a)}$ is then passed through a modality-specific encoder $E_m(\cdot)$ to produce a normalized embedding $\mathbf{E}_m^{(a)} = E_m(\tilde{\mathbf{X}}_m^{(a)})$, where $\mathbf{E}_m^{(a)} \in \mathbb{R}^E$. The resulting set of embeddings can then be aligned using a shared prototype space.

## 3.2 SHARED PROTOTYPE SPACE

To align representations across modalities, we introduce a set of $P$ trainable prototype vectors, organized as columns in a matrix $\mathbf{P} = [\mathbf{p}_1, \mathbf{p}_2, \ldots, \mathbf{p}_P] \in \mathbb{R}^{E \times P}$. This prototype space is shared across all modalities and training instances, serving as a common set of semantic anchors.

Each embedding $\boldsymbol{E}_m^{(a)}$ is projected onto the prototypes yielding similarity scores $\boldsymbol{S} \in \mathbb{R}^P$. To convert these scores into a soft assignment, we use two distinct transformations. First, we compute a probability distribution $\mathbf{U}_m^{(a)}$ using a softmax function with a temperature parameter $\tau$. Second, we compute an assignment target $\mathbf{V}_m^{(a)}$ using the Sinkhorn-Knopp algorithm (Cuturi, 2013), which enforces an equipartition constraint to prevent mode collapse by ensuring all prototypes are utilized equally across a batch.

$$\boldsymbol{S}_m^{(a)} = \boldsymbol{E}_m^{(a)} \boldsymbol{P} \tag{1}$$

$$\boldsymbol{U}_m^{(a)} = \text{Softmax}(\boldsymbol{S}_m^{(a)}/\tau) \tag{2}$$

$$\boldsymbol{V}_m^{(a)} = \text{Sinkhorn}(\boldsymbol{S}_m^{(a)}) \tag{3}$$

Now, the probability vector $\mathbf{U}$ and an assignment target $\mathbf{V}$ can be aligned via a cross entropy loss:

$$\ell(\boldsymbol{V}, \boldsymbol{U}) = -\boldsymbol{V} \cdot \log \boldsymbol{U} \tag{4}$$

## 3.3 MULTIMODAL PROTOTYPE PREDICTION LOSS

The key intuition of our method is that any pair of views originating from the same underlying system, regardless of modality, should be able to predict each other's prototype assignment. This is enforced through a swapped prediction loss comprising two components.

First, the within-modality loss, $\mathcal{L}_{\text{within-mod}}$, enforces consistency between all pairs of augmentations within a modality:

$$\mathcal{L}_{\text{within-mod}} = \sum_{m=1}^{M} \sum_{a=1}^{A} \sum_{b=1, b \neq a}^{A} \ell\left(\boldsymbol{U}_m^{(a)}, \boldsymbol{V}_m^{(b)}\right) \tag{5}$$

Second, the between-modality loss, $\mathcal{L}_{\text{between-mod}}$, enforces consistency between all pairs from different modalities:

$$\mathcal{L}_{\text{between-mod}} = \sum_{m=1}^{M} \sum_{n=1, n \neq m}^{M} \sum_{a=1}^{A} \sum_{b=1}^{A} \ell\left(\boldsymbol{U}_m^{(a)}, \boldsymbol{V}_n^{(b)}\right) \tag{6}$$

Our final objective, the Multimodal Prototype Prediction Loss ($\mathcal{L}_{\text{MPP}}$), is a linear combination of these two losses, normalized for stability:

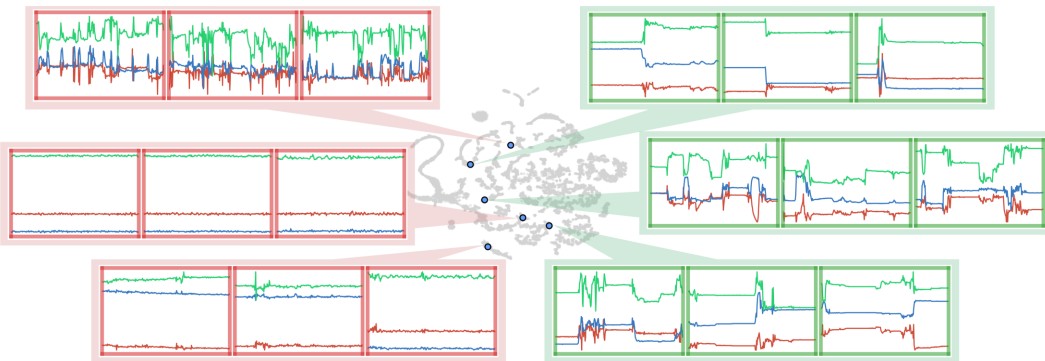

Figure 2: t-SNE of learned prototypes (gray), with k-means centroids (blue) and their top-three nearest **accelerometer** time-series. Panel borders denote ground-truth labels (green = Unstressed, red = Stressed). Each centroid captures a distinct motion motif, from active oscillatory bursts (top left) to sedentary plateaus (top right).

$$\mathcal{L}_{\text{MPP}} = \frac{1}{\binom{A \times M}{2}} (\alpha \mathcal{L}_{\text{within-mod}} + (1 - \alpha) \mathcal{L}_{\text{between-mod}}), \tag{7}$$

The hyperparameter $\alpha \in [0, 1]$ balances the contribution of the two objectives. Setting $\alpha = 1$ reduces the objective to independent, unimodal swapped prediction on each modality found in SwAV (Caron et al., 2020). In this work, we set $\alpha = 0.5$ to equally weight both sources of learning.

## 4 EXPERIMENTAL DESIGN

In Section 4.1, we first detail our large-scale pre-training dataset, along with the architecture and training settings. Then, in Section 4.2, we describe the unimodal and multimodal baselines that we compare against. Finally, in Section 4.3, we discuss our downstream datasets, their associated tasks, as well as our evaluation procedure.

### 4.1 PRE-TRAINING SETUP

Our pre-training data comprises the initial 10 days of data from the large-scale Mobile Open Observation of Daily Stressors (MOODS) study Neupane et al. (2024). It contains 794,872 synchronized 30-second segments of accelerometer and PPG signals from 122 participants (39 men, 77 women, 6 non-binary; mean age 38±13 years). The data was collected "in-the-wild" from a wrist-worn WearOS smartwatch as participants went about their daily lives without specific instructions. Further details on the study design are available in Neupane et al. (2024). To further demonstrate the broad applicability of our method across multimodal time-series domains, we also pre-train on the Moving Object Detection (MOD) dataset by Liu et al. (2023). It provides seismic vibration and audio recordings generated by different types of moving vehicles. We resample both modalities to 100 Hz and follow the original 8:1:1 split for training, validation, and testing. All segments are fixed at 2 seconds.

To ensure fair comparison, all models and modalities use the same encoder architecture: a 1D ResNet-26 with a kernel size of 11, a stride of 2, and a final embedding dimension of 512. The only architectural difference is the number of input channels for the initial convolution layer: 3 for accelerometer data, 1 for PPG, and 4 for the early-fusion models that concatenate inputs. In Table 1, the "Input" column denotes early-fusion models with concatenated inputs as P+A and multimodal models with separate encoders as P|A. We also use a consistent set of augmentations: for the accelerometer, we use an empirically-established set comprising additive Gaussian noise, scaling, 3D rotation, negation, time reversal, channel shuffle, segment shuffle, and time warping (Tang et al., 2020; Xu et al., 2024b). For the single-channel PPG signal, we use the same set, omitting the inapplicable rotation and channel shuffling augmentations.

We utilize the Adam optimizer (Kingma, 2014) with a learning rate of $10^{-5}$, no weight decay, and batch size of 256. Models are trained for 100 epochs or 96 hours (whichever comes first) on an NVIDIA L40S GPU. Both training and validation loss are calculated per batch but aggregated at epoch level, and the checkpoint with lowest validation loss is used for all downstream experiments.

## 4.2 BENCHMARKS

We compare ProtoMM against a comprehensive suite of multimodal and unimodal self-supervised baselines. Unless otherwise specified, all baselines are trained from scratch using the identical pre-training setup described above. Shared or modality-specific projection heads are incorporated following each baseline's original specifications. The baselines are as follows:

- **CLIP** (Radford et al., 2021; Thapa et al., 2024): Employs a contrastive loss to align the representations of temporally-paired between-modal signals.
- **COCOA** (Deldari et al., 2022): A multimodal contrastive method that aligns representations across modalities. The positive pairs are obtained from other temporally aligned sensors, whereas the negatives are mined from the same sensor, but from temporally misaligned data.
- **CroSSL** (Deldari et al., 2024): It stacks embeddings from modality-specific encoders, and performs random masking in the embedding space. A between-modal aggregator is utilized to obtain global embeddings, and the training is performed using the VICReg loss.
- **FOCAL** (Liu et al., 2023): A multimodal contrastive method that factorized representations into orthogonal between and within-modal subspaces, while enforcing temporal locality. Training in the frequency domain uses temporal and spectral augmentations to form within-modal positives, and temporally aligned sensors to form between-modal positives.
- **SLIP** (Mu et al., 2022): A straightforward multimodal method that combines SimCLR and CLIP objectives to learn within and between-modality interactions, respectively.
- **SimCLR** (Chen et al., 2020): A unimodal contrastive learning framework that utilizes augmented versions of input data. This is the unimodal version of SLIP.
- **ProtoMM Within-Mod**: A unimodal ablation of our method with $\alpha = 1$ that only performs within-modal. This is equivalent to applying SwAV (Caron et al., 2020).
- **RelCon** (Xu et al., 2024b): A unimodal contrastive learning method that leverages a learnable distance metric to assign soft positive pairs and enforce relative similarity ordering among all candidate segments.
- **Abbaspourazad et al. (2023)**: A unimodal contrastive-based framework that forms positive pairs from segments of the same subject and exploits a momentum-updated encoder to enrich representation diversity.

## 4.3 DOWNSTREAM EVALUATION SETUP

For evaluation, we use three datasets that contain synchronous PPG and accelerometer data, along with annotations: MOODS (stress detection, activity recognition) (Neupane et al., 2024), WESAD (stress detection) (Schmidt et al., 2018), and PPG-DaLiA (activity recognition, instantaneous heart rate prediction) (Reiss et al., 2019). All signals are resampled to 50 Hz to match the sampling frequency in pre-training dataset. After pre-training is complete, we freeze the model encoders and train a linear probe on the downstream tasks. For multimodal models, we generally will concatenate the embeddings from the PPG and accelerometer encoders to form the final representation. This is indicated by P+A in the "Out" column in our Results Table 1 and 2. For unimodal baselines, we use the single embedding directly. This is indicated by a P in the Out column in our Results Table 1.

- **MOODS** contains PPG and accelerometer signals collected using a Fossil Sport (Version 4) smartwatch from 122 participants. For stress detection (Stressed vs. Unstressed), we split the dataset into 1-minute windows following previous works (Mishra et al., 2018; Toshnazarov et al., 2024); whereas for binary activity recognition (Stationary vs. Non-stationary), we use 20-second samples for downstream evaluations.
- **WESAD** uses a wrist-worn Empatica E4 (McCarthy et al., 2016) to record accelerometer data (at 32 Hz) and blood volume pulse (at 64 Hz) from 15 participants. The stress detection task includes four sessions: Stress, Baseline, Amusement, and Meditation. For the binary classification task, following prior work (Schmidt et al., 2018; Dahal et al., 2023; Lange et al., 2024), we drop Meditation, merge Baseline and Amusement into Non-stress, while retaining the original Stress

Table 1: Linear-probe evaluation of unimodal and multimodal baselines. ProtoMM achieves best overall performance which indicates that grounding the alignment in a shared, discrete prototype space is a more effective mechanism for learning both shared and unique features simultaneously.

| | | | MOODS | | | | WESAD | | | | PPG-DaLiA | | | |
| | | | Stress (2) | | Activity (2) | | Stress (4) | | Stress (2) | | Activity (9) | | HR (R) | |
| Model | In | Out | ↑F1 | ↑Acc | ↑F1 | ↑Acc | ↑F1 | ↑Acc | ↑F1 | ↑Acc | ↑F1 | ↑Acc | ↓MAE | ↑$R^2$ |
|---|---|---|---|---|---|---|---|---|---|---|---|---|---|---|
| SimCLR | A | A | 0.477±0.006 | 0.598±0.006 | 0.861±0.004 | 0.925±0.002 | 0.557±0.053 | 0.629±0.052 | 0.793±0.055 | 0.836±0.044 | 0.559±0.005 | 0.560±0.004 | 15.37±0.10 | 0.101±0.015 |
| SimCLR | P | P | 0.527±0.006 | 0.607±0.006 | 0.655±0.005 | 0.857±0.002 | 0.348±0.034 | 0.517±0.053 | 0.692±0.056 | 0.699±0.054 | 0.302±0.004 | 0.399±0.004 | **8.14±0.07** | **0.670±0.008** |
| PMM WMod | A | A | 0.464±0.006 | 0.604±0.006 | 0.857±0.004 | 0.924±0.002 | 0.533±0.057 | 0.584±0.053 | 0.675±0.060 | 0.726±0.053 | 0.586±0.005 | 0.577±0.004 | 15.45±0.10 | 0.097±0.015 |
| PMM WMod | P | P | 0.469±0.005 | 0.611±0.005 | 0.595±0.005 | 0.848±0.003 | 0.522±0.055 | 0.607±0.053 | 0.847±0.050 | 0.877±0.039 | 0.285±0.004 | 0.397±0.004 | 8.29±0.07 | **0.670±0.007** |
| SimCLR | P+A | P+A | 0.488±0.006 | 0.601±0.006 | 0.849±0.004 | 0.919±0.002 | 0.445±0.048 | 0.596±0.053 | 0.721±0.056 | 0.753±0.051 | 0.542±0.005 | 0.536±0.004 | 9.91±0.08 | 0.534±0.009 |
| PMM WMod | P+A | P+A | 0.467±0.006 | 0.600±0.006 | 0.836±0.004 | 0.913±0.002 | 0.486±0.054 | 0.562±0.055 | 0.753±0.056 | 0.795±0.047 | 0.543±0.005 | 0.547±0.005 | 15.07±0.09 | 0.155±0.014 |
| Abbaspourazad et. al. | P+A | P+A | 0.445±0.005 | 0.614±0.005 | 0.681±0.005 | 0.865±0.002 | 0.417±0.044 | 0.551±0.056 | 0.668±0.062 | 0.753±0.050 | 0.444±0.005 | 0.453±0.004 | 15.38±0.10 | 0.103±0.014 |
| RelCon | P+A | P+A | 0.520±0.006 | 0.589±0.005 | 0.748±0.004 | 0.880±0.002 | 0.573±0.052 | 0.640±0.051 | **0.910±0.035** | **0.918±0.031** | 0.433±0.005 | 0.461±0.005 | 9.63±0.08 | 0.578±0.009 |
| CLIP | P\|A | P+A | 0.524±0.006 | 0.578±0.006 | 0.869±0.003 | 0.929±0.002 | 0.496±0.048 | 0.618±0.053 | **0.910±0.034** | **0.918±0.031** | 0.640±0.005 | 0.624±0.004 | 9.37±0.07 | 0.609±0.008 |
| COCOA | P\|A | P+A | 0.520±0.006 | 0.579±0.005 | 0.858±0.003 | 0.924±0.002 | 0.508±0.050 | 0.607±0.053 | 0.778±0.053 | 0.808±0.046 | 0.620±0.005 | 0.602±0.004 | 9.43±0.07 | 0.615±0.008 |
| CroSSL | P\|A | P+A | 0.461±0.006 | **0.622±0.006** | 0.751±0.004 | 0.882±0.002 | 0.430±0.045 | 0.494±0.052 | 0.783±0.053 | 0.808±0.047 | 0.553±0.005 | 0.540±0.004 | 11.47±0.08 | 0.464±0.010 |
| FOCAL | P\|A | P+A | 0.510±0.006 | 0.588±0.006 | 0.853±0.003 | 0.921±0.002 | 0.596±0.049 | 0.708±0.049 | 0.888±0.041 | 0.904±0.035 | 0.607±0.005 | 0.606±0.004 | 11.66±0.08 | 0.449±0.010 |
| SLIP | P\|A | P+A | 0.524±0.006 | 0.586±0.006 | 0.867±0.003 | 0.929±0.002 | 0.500±0.039 | 0.652±0.051 | 0.848±0.046 | 0.863±0.040 | 0.633±0.004 | 0.625±0.004 | 8.89±0.08 | 0.634±0.008 |
| **ProtoMM** | P\|A | P+A | **0.532±0.006** | 0.591±0.006 | **0.872±0.003** | **0.932±0.002** | **0.622±0.050** | **0.719±0.048** | **0.910±0.035** | **0.918±0.031** | **0.656±0.004** | **0.638±0.004** | 8.74±0.07 | 0.648±0.008 |

KEY: Encoder (In)put, Final Embedding (Out)put; (A)ccel, (P)PG; + designates concatenation, | designates separate encoders

sessions. Each session is segmented into non-overlapping 1-minute windows for downstream evaluations.

- **PPG-DaLiA** contains accelerometer data (at 32 Hz) and blood volume pulse (at 64 Hz), collected using an Empatiaca E4 (McCarthy et al., 2016) worn on the wrist and ECG (700 Hz) from chest-worn RespiBAN (biosignalsplux, 2019) to offer the ground truth of heart rate prediction. Fifteen subjects followed a semi-structured daily life protocol comprising eight distinct activities (Sitting, Ascending and Descending stairs, Table soccer, Cycling, Driving, Lunch break, Walking, and Working), with transient segments between activities annotated as an additional class. For model input, we segment all signals into 8-second windows with a 2-second sliding step.

We report macro F1-score and accuracy for classification tasks (stress, activity) and mean absolute error (MAE) and $R^2$ for the regression task (heart rate prediction).

## 5 RESULTS AND DISCUSSION

In this section we discuss four key findings. First, we clearly demonstrate that prototype-based approach improves upon the established contrastive learning setup. Next, we contrast ProtoMM against a full suite of multimodal benchmarks and demonstrate state-of-the-art performance. Then, we demonstrate how ProtoMM performs well because it is able to effectively integrate within- and between-modality information. Finally, we visualize the learned prototype space, and through examples show how the prototypes capture semantic meaning and specific morphologies.

**Prototypes Explicitly Improve Performance.** SLIP serves as the prototype-free analogue to ProtoMM. We utilize identical architecture and augmentation set for both models, making them equivalent up to the final embedding layer that produces $\boldsymbol{E}_m^{(a)}$. Then, ProtoMM projects these embeddings onto a learned set of prototypes before applying a swapped prediction loss across between- and within-modality pairs, but SLIP directly applies a contrastive (NT-Xent) loss to the embeddings themselves across all between- and within-modality pairs. Consequently, this comparison enables us to quantify the contribution of the prototype mechanism.

The results in Table 1 confirm the advantages of our prototype-based method. It demonstrates superior performance over its direct prototype-free analogue, SLIP, on every metric across all six downstream tasks. The performance gains are particularly pronounced on the WESAD dataset, where ProtoMM improves the F1-score for 4-class stress detection by a significant 24.6% (0.623 vs. 0.500) and for binary stress detection by over 7% (0.910 vs. 0.848).

Interestingly, performance gains remain isolated to the multimodal setting. In unimodal settings, SimCLR consistently outperforms its prototype-based counterpart, ProtoMM W-Mod. This aligns with prior work in unimodal time-series self-supervision (Meng et al., 2023). We hypothesize this is for two reasons: first, prototypes may act as a shared, discretized vocabulary that provides a common language to translate between disparate data streams like PPG and accelerometry. In a unimodal setting, the augmented views already reside on the same manifold, rendering this translation mechanism redundant. Second, the discrete bottleneck acts as a regularizer designed to filter out non-shared, modality-specific noise. In the unimodal case, where such independent noise sources are absent, this discretization acts effectively as a low-pass filter, discarding fine-grained continuous features that SimCLR preserves for richer instance discrimination.

Table 2: ProtoMM achieves the best performance at $\alpha$=0.5 and 0.67, showing that the unified MPP objective successfully captures both within- and between-modality information.

| Dataset | Task | Metric | $\alpha$ | | | | | |
| | | | 0 | 0.25 | 0.5 (ProtoMM) | 0.67 | 0.75 | 1 |
|---|---|---|---|---|---|---|---|---|
| MOODS | Stress (2) | ↑F1 | 0.461±0.005 | 0.463±0.005 | 0.532±0.006 | 0.522±0.006 | **0.537±0.006** | 0.497±0.006 |
| | | ↑Acc | 0.604±0.006 | **0.608±0.005** | 0.591±0.006 | 0.586±0.006 | 0.600±0.006 | 0.591±0.006 |
| | Activity (2) | ↑F1 | 0.719±0.005 | 0.724±0.005 | **0.872±0.003** | 0.869±0.003 | 0.871±0.003 | 0.855±0.004 |
| | | ↑Acc | 0.874±0.002 | 0.876±0.002 | **0.932±0.002** | 0.929±0.002 | 0.931±0.002 | 0.923±0.002 |
| WESAD | Stress (4) | ↑F1 | 0.578±0.057 | 0.589±0.058 | **0.622±0.050** | 0.577±0.054 | 0.526±0.047 | 0.612±0.055 |
| | | ↑Acc | 0.663±0.050 | 0.685±0.052 | **0.719±0.048** | 0.674±0.050 | 0.640±0.051 | 0.685±0.050 |
| | Stress (2) | ↑F1 | 0.783±0.052 | 0.786±0.054 | 0.910±0.035 | **0.938±0.031** | 0.800±0.051 | 0.858±0.049 |
| | | ↑Acc | 0.808±0.045 | 0.822±0.044 | 0.918±0.031 | **0.945±0.027** | 0.822±0.045 | 0.890±0.037 |
| PPG-DaLiA | Activity (9) | ↑F1 | 0.496±0.005 | 0.485±0.005 | 0.656±0.004 | 0.657±0.004 | **0.662±0.004** | 0.618±0.005 |
| | | ↑Acc | 0.502±0.004 | 0.493±0.004 | 0.638±0.004 | **0.651±0.004** | 0.648±0.004 | 0.603±0.004 |
| | HR (R) | ↓MAE | 10.45±0.08 | 10.76±0.08 | 8.74±0.07 | **8.67±0.07** | 9.08±0.08 | 9.14±0.08 |
| | | ↑$R^2$ | 0.536±0.009 | 0.515±0.009 | 0.648±0.008 | **0.654±0.007** | 0.618±0.008 | 0.611±0.008 |

**ProtoMM Achieves State-of-the-art Performance.** Table 1 presents comparisons against all baseline methods, which show that ProtoMM achieves SOTA results. It achieves the best overall performance in 4/6 downstream tasks and outperforms all other multimodal methods in 5/6 tasks.

The results show a clear trend where multimodal models outperform their unimodal counterparts, particularly on complex classification tasks. This highlights the value of alignment: one modality provides essential context that the other lacks. For example, heart rate information from PPG can help disambiguate activities with similar motion profiles from accelerometry, such as ascending versus descending stairs classes in PPG DaLiA activity classification or how research has shown that accelerometry and PPG signals can be used together for stress prediction (Sevil et al., 2020; Wu et al., 2015). The one exception to this trend is in the HR regression tasks. This highlights a key nuance in multimodal learning. Heart Rate is a metric that can be derived directly from the raw PPG waveform, and accelerometry signal gives little to no information on the heart rate. Therefore, the inclusion of accelerometry within the embedding model only serves to further obfuscate the final embedding, such that multimodal models generally do worse. However, ProtoMM is able to more effectively disentangle the modalities, to better preserve within-modal PPG-specific information, achieving the best multimodal performance.

Out of the multimodal baselines, CLIP, COCA, and CroSSL focusing on modeling only the between-modal information, whereas SLIP, FOCAL and our ProtoMM method are the ones that explicitly model both between and within-modal information. However, interestingly, the models that model both do not uniformly outperform the models that model only between-modal information. Despite SLIP augmenting the CLIP objective with an additional within-modal loss, their overall performance is comparable as the 2nd best models after ProtoMM. This suggests that standard contrastive losses may struggle to balance the two objectives, potentially over-emphasizing the more difficult between-modal alignment. ProtoMM's success indicates that grounding the alignment in a shared, discrete prototype space is a more effective mechanism for learning both shared and unique features simultaneously.

Finally, the table shows the early fusion models that concatenate the raw signals before encoder input perform poorly. Within Table 1, they are marked as P+A in the In column. This finding demonstrates that despite both being biosignals, a naive concatenation of PPG and accelerometry is insufficient, and modality-specific encoders are essential for learning how to capture PPG-specific and Accelerometry-specific features from the raw sensor data.

**Balancing within- and between-modal objectives.** The core advantage of ProtoMM lies in its ability to effectively simultaneously and effectively capture between- and within-modality information through a unified objective, $\mathcal{L}_{\text{MPP}}$. We validate this design choice by modifying the loss weighting parameter to be $\alpha = 1$, such that only within-modality ($\mathcal{L}_{\text{within-mod}}$) information is learned or to be $\alpha = 0$, such that only between-modality ($\mathcal{L}_{\text{between-mod}}$) information is learned.

Table 2 shows that the optimal performance is achieved at $\alpha = 0.5$, demonstrating that successful multimodal integration requires explicitly modeling both the unique contributions of each modality and their synergistic interactions. The balanced objective enables ProtoMM to learn representa-

Table 3: Integrating multimodal information into a unimodal embedding improves performance. ProtoMM rows show the performance of an unimodal embedding that was pre-trained multimodally (with both PPG and Accelerometer), while ProtoMM Within-Mod rows show the performance of an embedding pre-trained in isolation on just that single sensor.

| | | | | MOODS | | | | WESAD | | | | PPG-DaLiA | | | |
| | | | | Stress (2) | | Acitivity (2) | | Stress (4) | | Stress (2) | | Activity (9) | | HR (R) | |
| $\alpha$ | Model | In | Out | ↑F1 | ↑Acc | ↑F1 | ↑Acc | ↑F1 | ↑Acc | ↑F1 | ↑Acc | ↑F1 | ↑Acc | ↓MAE | ↑$R^2$ |
|---|---|---|---|---|---|---|---|---|---|---|---|---|---|---|---|
| .5 | **ProtoMM** | P\|A | A | **0.496** | 0.597 | **0.872** | **0.931** | **0.620** | **0.663** | **0.837** | **0.863** | **0.615** | **0.597** | **15.0** | **0.152** |
| 1 | PMM WMod | A | A | 0.464 | **0.604** | 0.857 | 0.924 | 0.533 | 0.584 | 0.675 | 0.726 | 0.586 | 0.577 | 15.5 | 0.097 |
| .5 | **ProtoMM** | P\|A | P | **0.541** | **0.613** | **0.747** | **0.882** | 0.518 | **0.607** | **0.855** | 0.863 | **0.332** | **0.426** | **8.16** | **0.686** |
| 1 | PMM WMod | P | P | 0.470 | 0.611 | 0.595 | 0.848 | **0.522** | **0.607** | 0.847 | **0.877** | 0.285 | 0.397 | 8.29 | 0.670 |

KEY: Encoder (In)put, Final Embedding (Out)put; (A)ccel, (P)PG; + designates concatenation, | designates separate encoders

Table 4: Finetuning with linear classifier evaluation of unimodal and multimodal baselines.

| | | | MOODS | | | | WESAD | | | | PPG-DaLiA | | | |
| | | | Stress (2) | | Activity (2) | | Stress (4) | | Stress (2) | | Activity (9) | | HR (R) | |
| Model | In | Out | ↑F1 | ↑Acc | ↑F1 | ↑Acc | ↑F1 | ↑Acc | ↑F1 | ↑Acc | ↑F1 | ↑Acc | ↓MAE | ↑$R^2$ |
|---|---|---|---|---|---|---|---|---|---|---|---|---|---|---|---|
| SimCLR | A | A | 0.388±0.002 | **0.634±0.005** | 0.859±0.003 | 0.925±0.002 | 0.468±0.052 | 0.528±0.053 | 0.681±0.064 | 0.767±0.050 | 0.587±0.004 | 0.586±0.004 | 18.33±0.11 | -0.196±0.010 |
| SimCLR | P | P | 0.388±0.002 | **0.634±0.005** | 0.737±0.005 | 0.879±0.002 | 0.504±0.040 | 0.618±0.051 | 0.807±0.049 | 0.822±0.045 | 0.332±0.005 | 0.409±0.004 | 17.92±0.11 | -0.175±0.009 |
| PMM WMod | A | A | **0.404±0.003** | 0.626±0.005 | 0.844±0.004 | 0.922±0.002 | 0.472±0.053 | 0.528±0.055 | 0.793±0.055 | 0.836±0.043 | 0.609±0.005 | 0.578±0.004 | **17.54±0.11** | **-0.122±0.008** |
| PMM WMod | P | P | 0.390±0.002 | **0.634±0.005** | 0.712±0.005 | 0.873±0.002 | 0.496±0.046 | 0.517±0.053 | **0.890±0.041** | **0.904±0.035** | 0.324±0.004 | 0.442±0.005 | 18.70±0.11 | -0.235±0.011 |
| SimCLR | P+A | P+A | 0.388±0.002 | **0.634±0.005** | 0.861±0.004 | 0.925±0.002 | **0.631±0.054** | **0.730±0.050** | 0.859±0.044 | 0.877±0.037 | 0.594±0.004 | 0.583±0.004 | 18.63±0.11 | -0.225±0.011 |
| PMM WMod | P+A | P+A | **0.404±0.003** | 0.627±0.005 | 0.859±0.004 | 0.926±0.002 | 0.485±0.053 | 0.562±0.054 | 0.793±0.055 | 0.836±0.043 | 0.626±0.004 | 0.582±0.004 | 18.77±0.11 | -0.241±0.012 |
| Abbaspourazad et. al. | P+A | P+A | 0.388±0.002 | **0.634±0.005** | 0.850±0.004 | 0.925±0.002 | 0.253±0.042 | 0.427±0.055 | 0.568±0.063 | 0.685±0.055 | 0.580±0.005 | 0.567±0.004 | 18.32±0.11 | -0.197±0.010 |
| RelCon | P+A | P+A | 0.388±0.002 | **0.634±0.005** | 0.860±0.003 | **0.927±0.002** | 0.512±0.054 | 0.596±0.053 | 0.706±0.063 | 0.795±0.045 | 0.625±0.005 | 0.592±0.004 | 18.00±0.11 | -0.165±0.010 |
| CLIP | P\|A | P+A | 0.388±0.002 | **0.634±0.005** | **0.867±0.003** | **0.929±0.002** | 0.518±0.052 | 0.618±0.053 | 0.815±0.046 | 0.822±0.045 | 0.554±0.004 | 0.570±0.004 | 17.66±0.11 | -0.143±0.008 |
| COCOA | P\|A | P+A | 0.388±0.002 | **0.634±0.005** | 0.859±0.003 | 0.926±0.002 | 0.575±0.048 | 0.652±0.051 | **0.935±0.031** | **0.945±0.026** | 0.540±0.005 | 0.565±0.005 | 18.45±0.11 | -0.210±0.011 |
| CroSSL | P\|A | P+A | 0.388±0.002 | **0.634±0.005** | 0.843±0.004 | 0.922±0.002 | 0.306±0.046 | 0.472±0.057 | 0.568±0.069 | 0.740±0.053 | 0.618±0.004 | **0.603±0.004** | 18.46±0.11 | -0.209±0.011 |
| FOCAL | P\|A | P+A | 0.388±0.002 | **0.634±0.005** | 0.849±0.004 | 0.922±0.002 | 0.402±0.042 | 0.562±0.053 | 0.611±0.066 | 0.740±0.050 | 0.618±0.005 | 0.599±0.004 | 18.42±0.11 | -0.202±0.011 |
| SLIP | P\|A | P+A | 0.390±0.002 | **0.634±0.005** | 0.852±0.004 | 0.926±0.002 | 0.522±0.039 | 0.640±0.052 | 0.862±0.046 | 0.890±0.035 | 0.534±0.005 | 0.509±0.004 | 19.28±0.11 | -0.282±0.013 |
| **ProtoMM** | P\|A | P+A | 0.388±0.002 | **0.634±0.005** | 0.857±0.004 | 0.921±0.002 | 0.559±0.047 | 0.652±0.053 | 0.888±0.041 | **0.904±0.034** | **0.665±0.004** | **0.622±0.004** | 18.38±0.11 | -0.201±0.011 |

KEY: Encoder (In)put, Final Embedding (Out)put; (A)ccel, (P)PG; + designates concatenation, | designates separate encoders

tions that are simultaneously invariant to modality-specific augmentations while being semantically aligned across modalities.

**Between-modal Knowledge Transfer to the Other Modality.** We would like to further explore how effective ProtoMM integrates Between-modal information by training ProtoMM normally, to capture both between-modal and within-modal information, and then instead of concatenating the outputs of each modality's embedding for evaluation, instead just using one modality's embedding at a time, show by the 1st and 3rd rows in Table 3. This will help us investigate whether the modality's trained embedding captures more information than if it was trained independently. Therefore, the baseline is training each modality independently with a unimodal ProtoMM Within-Mod ($\alpha = 1$) with only one modality as input, shown by the 2nd and 4th rows in Table 3.

As shown in Table 3, both encoders (i.e., for the accelerometer and PPG) outperform their unimodal ProtoMM Within-Mod counterparts on nearly all tasks, despite being evaluated without the full multimodal embedding. This indicates that the shared prototype space encourages each encoder to develop representations that are semantically aligned with the broader physiological context across modalities, not just its own signal characteristics, which results in more robust and informative features even when deployed as a unimodal embedding.

**Interpreting Prototypes.** To demonstrate that the explicit nature of our prototype-based learning leads to improved interpretability, we conduct a qualitative analysis on the WESAD stress detection dataset. We cluster the learned prototypes with k-means clustering (k=15), then for a given learned prototype centroid, we identify the accelerometer and PPG segments with highest cosine similarity in the representation space. Figure 2 and 3 shows a t-SNE projection of all prototypes, with representative examples highlighted alongside their top-3 nearest neighbors.

These neighbors exhibit both label consistency (stressed vs. unstressed) and coherent temporal dynamics that correspond to distinct physiological patterns (e.g., stable low-amplitude patterns, sharp oscillations, near-static segments with abrupt changes, strong trapezoid-like movements). This suggests that the shared prototype vectors are structured in the representation space in clusters that exhibit both label consistency and coherent physiological patterns. Rather than operating as a black box, ProtoMM's prototypes function as semantically meaningful anchors that capture both morphological signal characteristics and higher-level physiological contexts, offering a tangible insights into the latent physiological state.

Table 5: Evaluation of models pretrained on MOD dataset across evaluation protocols.

| Method | Metric | CLIP | COCOA | CroSSL | FOCAL | SLIP | ProtoMM |
|---|---|---|---|---|---|---|---|
| Linear-probe | ↑F1 | 0.702±0.012 | 0.220±0.008 | 0.702±0.012 | 0.639±0.013 | 0.737±0.011 | **0.782±0.011** |
| | ↑Acc | 0.719±0.011 | 0.306±0.012 | 0.719±0.011 | 0.652±0.012 | 0.749±0.011 | **0.787±0.011** |
| KNN | ↑F1 | 0.315±0.011 | 0.265±0.012 | 0.262±0.012 | 0.318±0.012 | 0.533±0.013 | **0.564±0.012** |
| | ↑Acc | 0.334±0.012 | 0.288±0.012 | 0.286±0.012 | 0.330±0.012 | 0.517±0.013 | **0.574±0.012** |
| Clustering | ↑ARI | 0.086±0.009 | 0.093±0.009 | 0.070±0.008 | 0.035±0.013 | **0.190±0.018** | 0.188±0.032 |
| | ↑NMI | 0.158±0.012 | 0.158±0.010 | 0.140±0.011 | 0.052±0.016 | 0.285±0.017 | **0.298±0.033** |

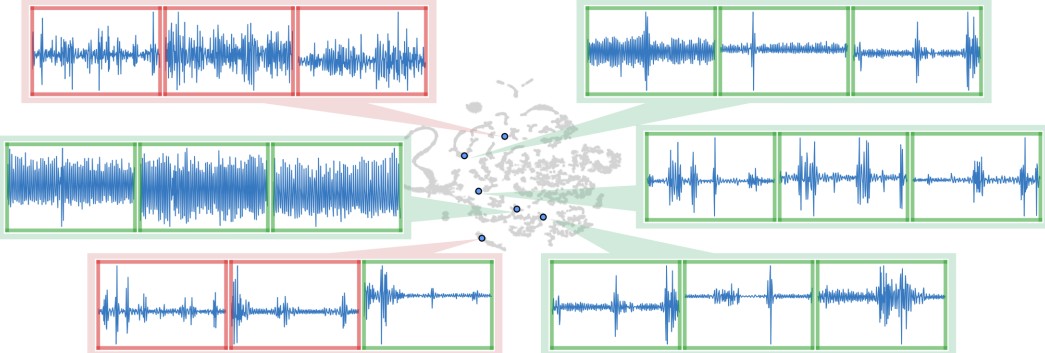

Figure 3: t-SNE of learned prototypes (gray), with k-means centroids (blue) and their top-three nearest **PPG** time-series. Panel borders denote ground-truth labels (green = Unstressed, red = Stressed). Each centroid captures a distinct pattern, from waveforms with high amplitude and variance (middle left) to those with a steady baseline and spiky variances (bottom right).

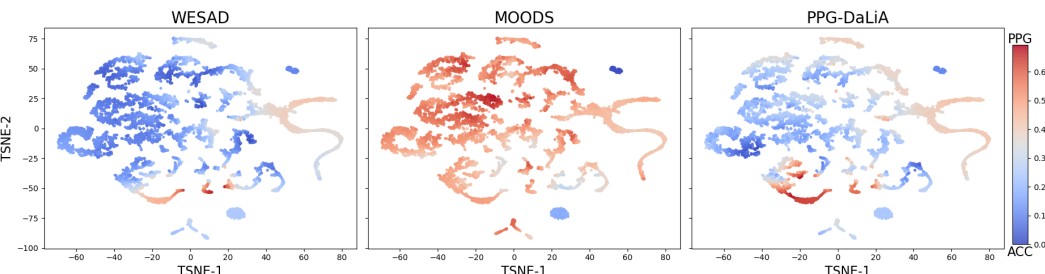

Figure 4: Across WESAD, MOODS, and PPG-DaLiA, t-SNE visualization of the learned prototype space, colored by the modality preference score. Only a smaller number of prototypes form consistent red (lower middle) or blue (upper right) subclusters across datasets, corresponding to modality-specific prototypes. The majority of prototypes exhibit varying colors across datasets, indicating that most prototypes capture shared latent states. This demonstrates that ProtoMM learns a hybrid representation consisting of mostly shared prototypes, while automatically discovering prototypes that encode complementary modality-specific patterns.

**Existence of modality-specific and modality-shared prototypes.** To quantify how strongly each prototype is activated by different sensor modalities, we compute a *modality preference score*. For each prototype vector $\mathbf{p}_k \in \mathbb{R}^E$, we evaluate its average assignment probability under each modality samples separately. For a given dataset, this probability assignment is calculated as,

$$\mathbf{p}_k^m = \mathbb{E}_i[\mathbf{U}_{m,i}[k]] \in \mathbb{R},$$

where the expectation is taken over all samples $i$ in modality $m$. We define the modality preference score as the difference between these assignment probabilities. For a PPG and Accelerometry dataset, the score of prototype $\mathbf{p}_k$ is shown as:

$$s_k = \mathbf{p}_k^{\text{PPG}} - \mathbf{p}_k^{\text{ACC}},$$

where strong positive scores correspond to PPG-specific prototypes, strong negative scores correspond to ACC-specific prototypes, and values near zero indicate modality-shared prototypes.

## 6 REPRODUCIBILITY

Our Methods section in Section 3 presents the model and training setup, and Experiments section in Section 4 describes the evaluation protocol and study design. Hyperparameters for every benchmark appear in Appendix A.2. We will release our model weights and a codebase with the full training methodology, architecture, and reproducible evaluation code, upon acceptance. The PPG-DaLiA and WESAD datasets are publicly available, and Section 4.3 explains how we curate and preprocess each one for our tasks.

## 7 ETHICS

Our paper develops models using physiological signals, with the goal of improving personal health. We acknowledge the associated risks, including privacy issues and the possibility of widening health disparities, as these models enable more detailed characterization of patients. Without effective regulation, patients may have limited control over their data, raising concerns about the upholding of autonomy, a core principle of medical ethics. Our study uses de-identified data from IRB-approved protocols, ensuring no participant identification information is included in our analysis. Nevertheless, we believe our work contributes positively to the field by advancing personalized health recommendations, which can enhance care quality, patient safety, and overall well-being. Additionally, we acknowledge the use of LLMs to assist in editing and polishing the writing for this submission, specifically, to edit phrasing and to clarify the framing of ideas in a manner that reflect the authors' original intent.

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

## A    APPENDIX

### A.1    LIMITATIONS

While ProtoMM effectively aligns multimodal biosignals, it relies on the assumption that physiological dynamics can be discretized into a finite set of shared prototypes. This design choice imposes a specific inductive bias that works well for quasi-periodic signals (e.g., cardiac cycles in PPG, gait in accelerometry) but may be suboptimal for highly stochastic or non-stationary signals lacking recurring motifs. Additionally, our shared prototype constraint assumes a symmetric semantic overlap

between modalities. In scenarios where one modality contains significant unique information irrelevant to the other (e.g., motion artifacts in accelerometry unrelated to heart rate), the strict alignment objective may suppress these modality-specific features if the prototype size is not sufficiently large to accommodate them.

## A.2 BENCHMARK IMPLEMENTATIONS

Our code will be made publicly available upon acceptance. All benchmark implementations adhere to the experimental setup described in Section 4, utilizing identical 1D ResNet-26 encoders with global max temporal pooling to produce 512-dimensional embeddings. Each training sample is transformed into two stochastic views, and augmentations are sampled with equal probability. Training hyperparameters remain consistent across all methods as described in Section 4.1. Below we detail the implementation-specific parameters for each baseline.

- **CLIP** (Radford et al., 2021; Thapa et al., 2024): Pairwise contrastive alignment between temporally paired modalities. We employ modality-specific single-layer projection heads with same input and output dimensions (512). The temperature parameter is initialized to 1.0.

- **COCOA** (Deldari et al., 2022): We reimplement the original TensorFlow code (https://github.com/cruiseresearchgroup/COCOA) in PyTorch while maintaining architectural fidelity. Key parameters include temperature=0.1, scale_loss=1/32, and lambd=3.9e-3, window = 100. Modality-specific projectors consist of a flattening layer followed by a linear projection to maintain dimensional consistency during training. Due to the method's specific design, global pooling is disabled during training.

- **CroSSL** (Deldari et al., 2024): We reimplement the original TensorFlow code (https://github.com/Nokia-Bell-Labs/CroSSL) in PyTorch, which follows the original spatial masking approach with coverage=0.9. Embeddings from modality-specific encoders are processed through a shared projector network consists of several linear layers with ReLU activations in between, resulting in a final output dimension (proj_size) of 32.

- **FOCAL** (Liu et al., 2023): We adapt the official implementation (https://github.com/tomoyoshki/focal) to our codebase while preserving the core methodology. Modality-specific projectors consist of two linear layers with ReLU activation. Key parameters include temperature=0.5, sequence_length=4, and loss weights: shared contrastive=1, private contrastive=1, orthogonal=3, rank=5. Augmentations include standard temporal transformations followed by frequency-domain phase shifting.

- **SLIP** (Mu et al., 2022): We implement both within- and between-modality contrastive losses using the standard CLIP formulation with temperature initialized to 1.0.

- **SimCLR** (Chen et al., 2020): Unimodal contrastive learning with two augmented views per sample. temperature=0.1 is used for the contrastive loss.

- **RelCon** (Xu et al., 2024b) We use the official RelCon implementation (https://github.com/maxxu05/relcon) and train its motif-based distance module on our pre-training datasets (MOODS, MOD). Following the default setup of the authors, we retain key hyperparameters withinuser_cands=20 and tau=0.1.

- **Abbaspourazad et al. (2023)** We implement the positive pair selection and momentum training as directed in the paper with tau=0.04, momentum_tau=0.99, and koleo_lambda=0.1.

All implementations maintain fairness through consistent encoder architecture, data preprocessing, and evaluation protocols. Differences arise only in method-specific components as detailed above, ensuring meaningful comparison while preserving each approach's distinctive characteristics.

## A.3 ABLATION STUDY

### A.3.1 EFFECT OF NUMBER OF PROTOTYPES

**Set up.** We conduct ablation study with number of prototypes $P$ = [0.3k, 1k, 3k, 10k, 30k, 100k].

**Analysis.** In Table A.2 Across datasets, we observe stable performance when P is greater or equal to 1k. The performance gains for extremely large sets of prototypes (30k, 100k) are small (e.g. 1.1% improvement on MOODS stress F1, 1.2% improvement on WESAD binary stress F1). This confirms

Table 6: Effect of the number of prototypes ($P$) on downstream performance.

| Dataset | Task | Metric | Number of Prototypes ($P$) | | | | | |
|---|---|---|---|---|---|---|---|---|
| | | | 0.3k | 1k | 3k | 10k (ProtoMM) | 30k | 100k |
| MOODS | Stress (2) | ↑F1 | 0.448±0.005 | 0.532±0.006 | 0.532±0.006 | 0.532±0.006 | 0.529±0.006 | **0.543±0.006** |
| | | ↑Acc | **0.616±0.005** | 0.587±0.006 | 0.593±0.006 | 0.591±0.006 | 0.586±0.005 | 0.597±0.006 |
| | Activity (2) | ↑F1 | 0.619±0.005 | 0.871±0.003 | 0.868±0.003 | **0.872±0.003** | 0.871±0.003 | 0.869±0.003 |
| | | ↑Acc | 0.855±0.002 | 0.931±0.002 | 0.929±0.002 | **0.932±0.002** | 0.931±0.002 | 0.930±0.002 |
| WESAD | Stress (4) | ↑F1 | 0.518±0.059 | **0.641±0.053** | 0.556±0.036 | 0.622±0.050 | 0.546±0.049 | 0.607±0.053 |
| | | ↑Acc | 0.596±0.055 | **0.719±0.047** | 0.697±0.050 | **0.719±0.048** | 0.618±0.052 | 0.697±0.049 |
| | Stress (2) | ↑F1 | 0.712±0.059 | 0.910±0.035 | 0.905±0.037 | 0.910±0.035 | **0.922±0.034** | 0.824±0.048 |
| | | ↑Acc | 0.767±0.049 | 0.918±0.032 | 0.918±0.032 | 0.918±0.031 | **0.932±0.030** | 0.836±0.044 |
| PPG-DaLiA | Activity (9) | ↑F1 | 0.498±0.005 | **0.661±0.004** | 0.656±0.004 | 0.656±0.004 | 0.645±0.004 | **0.661±0.004** |
| | | ↑Acc | 0.503±0.004 | **0.645±0.004** | 0.634±0.004 | 0.638±0.004 | 0.626±0.004 | 0.641±0.004 |
| | HR (R) | ↓MAE | 10.58±0.08 | **8.59±0.07** | 8.84±0.07 | 8.74±0.07 | 8.82±0.07 | 8.78±0.07 |
| | | ↑$R^2$ | 0.529±0.009 | **0.662±0.007** | 0.641±0.008 | 0.648±0.008 | 0.638±0.008 | 0.643±0.008 |

Table 7: Ablation study on Sinkhorn-Knopp normalization.

| | MOODS | | | | WESAD | | | | PPG-DaLiA | | | |
|---|---|---|---|---|---|---|---|---|---|---|---|---|
| | Stress (2) | | Activity (2) | | Stress (4) | | Stress (2) | | Activity (9) | | HR (R) | |
| Setting | ↑F1 | ↑Acc | ↑F1 | ↑Acc | ↑F1 | ↑Acc | ↑F1 | ↑Acc | ↑F1 | ↑Acc | ↓MAE | ↑$R^2$ |
| w/ Sinkhorn (ProtoMM) | 0.532±0.006 | 0.591±0.006 | 0.872±0.003 | 0.932±0.002 | 0.622±0.050 | 0.719±0.048 | 0.910±0.035 | 0.918±0.031 | 0.656±0.004 | 0.638±0.004 | 8.74±0.07 | 0.648±0.008 |
| Sinkhorn → Softmax | -11.0% | +2.6% | -24.2% | -7.8% | -9.2% | -7.8% | -14.6% | -11.9% | -30.1% | -26.3% | +28.0% | -24.9% |
| w/o Sinkhorn | -11.6% | +1.1% | -11.9% | -4.5% | -17.0% | -15.6% | -21.8% | -16.4% | -26.9% | -22.4% | +25.6% | -23.6% |

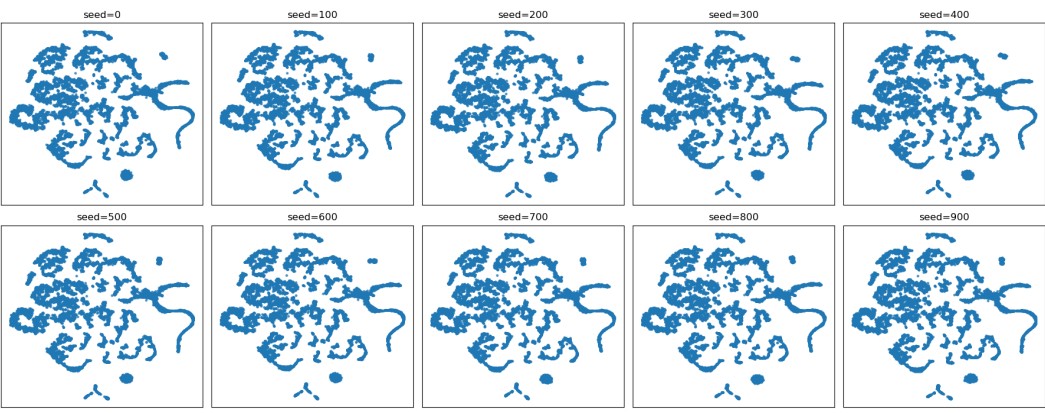

Figure 5: t-SNE visualization of prototype space across 10 different random seeds (0–900). The scatter patterns remain nearly identical, confirming that the learned prototypes form a stable structure independent of the initialization.

that ProtoMM does not rely on large prototype sets to function well. Overall, this demonstrates that ProtoMM is highly robust to prototype-set size.

### A.3.2 EFFECT OF SINKHORN-KNOPP NORMALIZATION

**Setup.** We investigate two ablations on the Sinkhorn-Knopp normalization in ProtoMM: ProtoMM **Sinkhorn → Softmax** which replaces Sinkhorn normalization with Softmax, and ProtoMM **w/o Sinkhorn** that disenables Sinkhorn by setting number of iterations as 0.

**Analysis** Table 7 presents the results of the ablation study. Both modifications yield large performance drops across datasets (e.g. PPG-DaLiA Activity F1 drops by 30.1% and 26.9%, respectively). This highlights that Sinkhorn-Knopp is a critical mechanism for avoiding collapse and ensuring stable alignment.

Table 8: Clustering evaluation.

| Dataset | Task | Metric | CLIP | COCOA | CroSSL | FOCAL | SLIP | ProtoMM |
|---|---|---|---|---|---|---|---|---|
| | | | | | Model | | | |
| MOODS | Stress (2) | ↑ARI | 0.003±0.003 | 0.004±0.002 | **0.005±0.003** | -0.002±0.001 | -0.001±0.002 | 0.003±0.002 |
| | | ↑NMI | 0.000±0.000 | **0.001±0.001** | 0.001±0.001 | 0.000±0.000 | 0.000±0.000 | 0.000±0.000 |
| | Activity (2) | ↑ARI | -0.033±0.003 | 0.002±0.002 | **0.006±0.005** | -0.032±0.003 | -0.075±0.003 | -0.057±0.023 |
| | | ↑NMI | 0.028±0.003 | 0.016±0.002 | 0.000±0.000 | 0.007±0.001 | **0.073±0.003** | 0.048±0.011 |
| WESAD | Stress (4) | ↑ARI | 0.188±0.075 | 0.140±0.074 | 0.029±0.036 | 0.212±0.077 | **0.218±0.070** | 0.190±0.091 |
| | | ↑NMI | 0.258±0.063 | 0.171±0.065 | 0.123±0.042 | 0.273±0.069 | **0.291±0.066** | 0.158±0.071 |
| | Stress (2) | ↑ARI | -0.012±0.075 | 0.113±0.106 | 0.170±0.091 | 0.088±0.102 | **0.262±0.105** | 0.198±0.124 |
| | | ↑NMI | 0.000±0.053 | 0.060±0.065 | **0.194±0.076** | 0.087±0.081 | 0.187±0.075 | 0.124±0.088 |
| PPG-DaLiA | Activity (9) | ↑ARI | 0.003±0.001 | 0.004±0.001 | -0.004±0.001 | **0.089±0.009** | 0.000±0.001 | 0.004±0.001 |
| | | ↑NMI | 0.014±0.002 | 0.012±0.001 | 0.006±0.001 | **0.171±0.009** | 0.002±0.001 | 0.026±0.002 |

Table 9: Evaluating the class purity of the prototypes: we retrieve the $k$ nearest neighbors for each prototype across modalities and analyze if the neighbors belong to the same class.

| Dataset | Task | Metric | CLIP | COCOA | CroSSL | FOCAL | SLIP | ProtoMM |
|---|---|---|---|---|---|---|---|---|
| | | | | | Model | | | |
| MOODS | Stress (2) | ↑F1 | 0.519±0.006 | 0.507±0.006 | 0.505±0.006 | 0.502±0.006 | 0.512±0.006 | **0.536±0.006** |
| | | ↑Acc | 0.550±0.006 | 0.537±0.006 | 0.549±0.006 | 0.550±0.006 | 0.545±0.006 | **0.563±0.006** |
| | Activity (2) | ↑F1 | 0.811±0.004 | 0.802±0.004 | 0.613±0.005 | 0.788±0.004 | 0.829±0.004 | **0.831±0.004** |
| | | ↑Acc | 0.903±0.002 | 0.897±0.002 | 0.830±0.003 | 0.893±0.002 | 0.911±0.002 | **0.914±0.002** |
| WESAD | Stress (4) | ↑F1 | 0.415±0.044 | 0.431±0.057 | 0.443±0.053 | **0.491±0.044** | 0.468±0.042 | 0.416±0.045 |
| | | ↑Acc | 0.539±0.053 | 0.539±0.056 | 0.528±0.054 | **0.607±0.056** | 0.562±0.053 | 0.528±0.055 |
| | Stress (2) | ↑F1 | 0.675±0.059 | 0.773±0.056 | 0.656±0.062 | **0.813±0.052** | 0.738±0.059 | 0.787±0.053 |
| | | ↑Acc | 0.726±0.051 | 0.822±0.044 | 0.740±0.050 | **0.849±0.041** | 0.795±0.047 | 0.836±0.042 |
| PPG-DaLiA | Activity (9) | ↑F1 | 0.235±0.005 | 0.191±0.004 | 0.162±0.004 | **0.457±0.005** | 0.153±0.004 | 0.236±0.005 |
| | | ↑Acc | 0.314±0.004 | 0.281±0.004 | 0.243±0.004 | **0.482±0.004** | 0.249±0.004 | 0.315±0.004 |

Table 10: Prototype class purity over binary classification tasks.

| Dataset | Task | Mod | Top-3 | Top-5 | Top-10 |
|---|---|---|---|---|---|
| | | | Top-$k$ Prototype Purity (%) | | |
| MOODS | Stress (2) | A | 41.95 | 23.92 | 9.21 |
| | | P | 36.56 | 16.33 | 4.56 |
| | Activity (2) | A | 83.57 | 77.65 | 70.13 |
| | | P | 79.80 | 71.16 | 61.15 |
| WESAD | Stress (2) | A | 53.74 | 33.12 | 13.26 |
| | | P | 64.85 | 45.62 | 27.38 |

KEY: (Mod)ality; (A)ccel, (P)PG

## A.4 DERIVATION OF OUR MULTIMODAL PROTOTYPE PREDICTION LOSS FROM LOG-LIKELIHOOD

Our objective is to find the network parameters $\theta$ that maximizes the log-likelihood function of the observed $i$ samples, composed of $M$ modalities. :

$$\theta^* = \arg\max_\theta \sum_i \sum_{a=1}^A \sum_{m=1}^M \log p(\mathbf{X}_{i,m}^{(a)}; \theta)$$

We assume that the observed data modality $\mathbf{X}_{i,m}^{(a)}$ is related to some combination of modality-specific and modality-shared prototypes $\mathbf{P} = \{\mathbf{P}_m \cup \mathbf{P}_s\}$ where $\mathbf{p}_{i,m}^{(a)}$ is the optimal prototype assignment of $\mathbf{X}_{i,m}^{(a)}$. We abbreviate $\mathbf{p}_{i,m}^{(a)}$ to $\mathbf{p}$ for brevity.

$$= \arg\max_\theta \sum_i \sum_{a=1}^A \sum_{m=1}^M \log \sum_{\mathbf{p} \in \{\mathbf{P}_m \cup \mathbf{P}_s\}} p(\mathbf{X}_{i,m}^{(a)}, \mathbf{p}; \theta)$$

Given a $Q(\mathbf{p})$ probability simplex for $\mathbf{p}$, such that $\sum_{\mathbf{p} \in \mathbf{P}} Q(\mathbf{p}) = 1$,

$$= \arg\max_\theta \sum_i \sum_{a=1}^A \sum_{m=1}^M \log \sum_{\mathbf{p} \in \{\mathbf{P}_m \cup \mathbf{P}_s\}} Q(\mathbf{p}) \frac{p(\mathbf{X}_{i,m}^{(a)}, \mathbf{p}; \theta)}{Q(\mathbf{p})}$$

With Jensen's inequality,

$$\geq \arg\max_\theta \sum_i \sum_{a=1}^A \sum_{m=1}^M \sum_{\mathbf{p} \in \{\mathbf{P}_m \cup \mathbf{P}_s\}} Q(\mathbf{p}) \log \frac{p(\mathbf{X}_{i,m}^{(a)}, \mathbf{p}; \theta)}{Q(\mathbf{p})}$$

Therefore, removing constants, we should maximize:

$$\sum_i \sum_{m=1}^M \sum_{\mathbf{p} \in \{\mathbf{P}_m \cup \mathbf{P}_s\}} Q(\mathbf{p}) \log p(\mathbf{X}_{i,m}^{(a)}, \mathbf{p}; \theta)$$

$$= \sum_i \sum_{m=1}^M \left( \sum_{\mathbf{p} \in \mathbf{P}_s} Q(\mathbf{p}) \log p(\mathbf{X}_{i,m}^{(a)}, \mathbf{p}; \theta) + \sum_{\mathbf{p} \in \mathbf{P}_m} Q(\mathbf{p}) \log p(\mathbf{X}_{i,m}^{(a)}, \mathbf{p}; \theta) \right)$$

Now with $\mathbf{U}_m^{(a)} = \mathrm{Softmax}(\boldsymbol{E}_m^{(a)}\boldsymbol{P}/\tau)$, the softmax forms a probability distribution over $\mathbf{X}_{i,m}^{(a)}$ and $\mathbf{p}$, so we can substitute it in.

$$= \sum_i \sum_{m=1}^M \sum_{a=1}^A \left( \sum_{\mathbf{p} \in \mathbf{P}_s} Q(\mathbf{p}) \log \mathbf{U}_m^{(a)} + \sum_{\mathbf{p} \in \mathbf{P}_m} Q(\mathbf{p}) \log \mathbf{U}_m^{(a)} \right)$$

Now with $\boldsymbol{V}_m^{(a)} = \mathrm{Sinkhorn}(\boldsymbol{E}_m^{(a)}\boldsymbol{P})$, the sinkhorn give us normalized soft prototype label assignments, we can substitute them in as being proportional to a true probability simplex.

$$\propto \sum_i \sum_{m,n=1}^M \sum_{a,b=1}^A \left( \sum_{n \neq m} \mathbf{V}_n^{(a)} \log \mathbf{U}_m^{(a)} + \sum_{b \neq a} \mathbf{V}_m^{(b)} \log \mathbf{U}_m^{(a)} \right) = -\mathcal{L}_{\mathrm{MPP}} * \beta$$

This is now equivalent our negative loss function multiplied by a constant $\beta$. Therefore, by minimizing our $\mathcal{L}_{\mathrm{MPP}}$, we are maximizing this lower bound of the log-likelihood.

