# OpenReview forum: "Leveraging Shared Prototypes for a Multimodal Pulse Motion Foundation Model"
_ICLR.cc/2026/Conference — Submitted to ICLR 2026_

### Official Review · Reviewer_1JH9 · 2025-10-30

**Soundness:** 2
**Presentation:** 2
**Contribution:** 2
**Rating:** 2
**Confidence:** 2

**Summary:**

The paper presents ProtoMM, a prototype-based self-supervised learning method for biosignals (ECG, PPG). ProtoMM aligns views of one sample of the same and different modalities, exploiting within- and between-modality information.

**Strengths:**

- The paper is well-written and easy to follow
- The experimental setup is clear showing the advantages of the method compared to the baselines

**Weaknesses:**

1/ Outdated baselines.

1a/ Foundation models: The main weakness of the work is the fact that the baselines are outdated. There exist several modern works that use foundation models for biosignals; in all cases the procedure is different to the one proposed in this work, but we cannot understand how well the proposed method works without modern baselines.

[a] Xu et. al.,  RelCon: Relative Contrastive Learning for a Motion Foundation Model for Wearable Data, ICLR 2025
[b] Abbaspourazad, et. al., Large-scale Training of Foundation Models for Wearable Biosignals, ICLR 2024
[c] Narayanswamy et. al., Scaling wearable foundation models, ICLR 2025
[d] Saha et. al., Pulse-PPG: An Open-Source Field-Trained PPG Foundation Model for Wearable Applications across Lab and Field Settings, ACM on Interactive, Mobile, Wearable and Ubiquitous Technologies

1b/ Self supervised methods:
There exist several more recent SSL methods operating on biosignals that could be explored or adapted. See some examples below:

[e] Tag et. al., Electrocardiogram Report Generation and Question Answering via Retrieval-Augmented Self-Supervised Modeling, ICASSP 2025
[f] Mordacq et. al., ADAPT: Multimodal Learning for Detecting Physiological Changes under Missing Modalities, MIDL 2024
[g] Shen et. al., CIMSleepNet: Robust Sleep Staging over Incomplete Multimodal Physiological Signals via Contrastive Imagination, NeurIPS 2024

Given the 1a/ and 1b/ categories, right now the proposed method seems standalone without proper bibliographic discussion and without convincing baselines and comparison of other methods.

2/ Modalities.
The paper has evaluated only two modalities, which makes claims such as “general multimodal framework” not convincing. It would be useful to have at least one higher-level modality (e.g., EDA, ECG, or text).

3/ Datasets.
The datasets seem outdated and it seems that there exist newer and more within scope datasets:
[i] EEVR: A Dataset of Paired Physiological Signals and Textual Descriptions for Joint Emotion Representation Learning, NeurIPS 2024
[ii] WildPPG (A Real-World PPG Dataset of Long Continuous Recordings), NeurIPS 2024
[iii] Stressid: a multimodal dataset for stress identification, NeurIPS 2023

4/ Methodology
While intuitively well-motivated, the paper lacks a deeper theoretical justification of why prototype consistency across modalities results in better disentanglement or regularization. Furthermore, there is no discussion on the convergence or stability of the Sinkhorn-based assignment.

5/ Missing ablations
- Several ablations are missing (hyperparam robustnes, number of prototypes, temperature, choice of \alpha, etc). These could really help understand what works and why in the proposed method.
- It remains unclear whether ProtoMM embeddings are better than contrastive baselines so a baseline with contrastive learning would help..

6/ Frozen encoders.
All downstream evaluations use frozen encoders with linear probes thus making the method dependent on the quality of the encoder.

7/ Visualizations and analysis.
Although there exist some visualizations, we cannot understand the importance of training with both losses neither the importance of prototypes. It would have been helpful to have these. For example, we cannot understand if the shared prototype space is truly modality-agnostic, or if some modality-specific subclusters have emerged.

8/ Prototypes.
Prototypes are the main contribution of the work but we cannot understand how they operate. Besides visualization, we would now what happens for instance when we have one dominant modality or how many we need to have or if we need additional latents to represent them or what would happen if we have longer sequences. Analysing them and having findings transferable to larger datasets and other modalities would strengthen the work.

9/ Failure cases are not discussed.

**Questions:**

It would have been useful to have answers to the weaknesses above.
Some additional questions are:

Q1: it would be interesting to have the computational overhead of the Sinkhorn-based prototype updates compared to contrastive methods

Q2 (W6) How does ProtoMM perform when fine-tuned end-to-end on downstream tasks, compared to linear probing?

Q3 Are the learned prototypes stable across training runs, or do they vary significantly with initialization?

---

> ### Author Response · Authors · 2025-11-26
>
> # 1. Baselines
> ### Addressing:
> * W1a: “The main weakness of the work is the fact that the baselines are outdated. There exist several modern works that use foundation models for biosignals; in all cases the procedure is different to the one proposed in this work, but we cannot understand how well the proposed method works without modern baselines.”
> * W1b: “There exist several more recent SSL methods operating on biosignals that could be explored or adapted”
>
> ### Response:
> Thank you for your great suggestions. We have added RelCon and Abbaspourazad et al. as our baselines. These are unimodal approaches, but we include them in a multimodal fashion via an early fusion approach. As our paper is method-focused, we re-train RelCon and Abbaspourazad et. al. on our dataset to ensure a fair and controlled comparison to ProtoMM.
>
> Table 1 in the paper (also added below) now contains an expanded list of baselines. Still, ProtoMM is the best multimodal method overall. RelCon and Abbaspourazad et al. are not effective in the multi-modal setting.
>
> To ensure a fair comparison, all models share identical data preprocessing, encoder architectures, training hyperparameters, and evaluation protocols. The only differences stem from method-specific components, for which we adopt the hyperparameters reported in the original papers. For detailed information, please refer to Section 4 and A.2. Our results demonstrate that ProtoMM is the best multimodal method (see Table 1 in the paper for reference).
>
> **Table 1: Results on MOODS Dataset**
>
> | Model | In | Out | Stress (2) F1 (↑) | Stress (2) Acc (↑) | Activity (2) F1 (↑) | Activity (2) Acc (↑) |
> | :--- | :---: | :---: | :---: | :---: | :---: | :---: |
> | **Unimodal** | | | | | | |
> | SimCLR | A | A | 0.477±0.006 | 0.598±0.006 | 0.861±0.004 | 0.925±0.002 |
> | SimCLR | P | P | _0.527±0.006_ | 0.607±0.006 | 0.655±0.005 | 0.857±0.002 |
> | PMM WMod | A | A | 0.464±0.006 | 0.604±0.006 | 0.857±0.004 | 0.924±0.002 |
> | PMM WMod | P | P | 0.469±0.005 | 0.611±0.005 | 0.595±0.005 | 0.848±0.003 |
> | **Multimodal** | | | | | | |
> | SimCLR | P+A | P+A | 0.488±0.006 | 0.601±0.006 | 0.849±0.004 | 0.919±0.002 |
> | PMM WMod | P+A | P+A | 0.467±0.006 | 0.600±0.006 | 0.836±0.004 | 0.913±0.002 |
> | [NEW] Abbaspourazad et al. | P+A | P+A | 0.445±0.005 | _0.614±0.005_ | 0.681±0.005 | 0.865±0.002 |
> | [NEW]  RelCon | P+A | P+A | 0.520±0.006 | 0.589±0.005 | 0.748±0.004 | 0.880±0.002 |
> | CLIP | P\|A | P+A | 0.524±0.006 | 0.578±0.006 | _0.869±0.003_ | _0.929±0.002_ |
> | COCOA | P\|A | P+A | 0.520±0.006 | 0.579±0.005 | 0.858±0.003 | 0.924±0.002 |
> | CroSSL | P\|A | P+A | 0.461±0.005 | **0.622±0.006** | 0.751±0.004 | 0.882±0.002 |
> | FOCAL | P\|A | P+A | 0.510±0.006 | 0.588±0.006 | 0.853±0.003 | 0.921±0.002 |
> | SLIP | P\|A | P+A | 0.524±0.006 | 0.586±0.006 | 0.867±0.003 | _0.929±0.002_ |
> | **ProtoMM** | P\|A | P+A | **0.532±0.006** | 0.591±0.006 | **0.872±0.003** | **0.932±0.002** |
>
> **Table 2: Results on WESAD Dataset**
>
> | Model | In | Out | Stress (4) F1 (↑) | Stress (4) Acc (↑) | Stress (2) F1 (↑) | Stress (2) Acc (↑) |
> | :--- | :---: | :---: | :---: | :---: | :---: | :---: |
> | **Unimodal** | | | | | | |
> | SimCLR | A | A | 0.557±0.053 | 0.629±0.052 | 0.793±0.055 | 0.836±0.044 |
> | SimCLR | P | P | 0.348±0.034 | 0.517±0.053 | 0.692±0.056 | 0.699±0.054 |
> | PMM WMod | A | A | 0.533±0.057 | 0.584±0.053 | 0.675±0.060 | 0.726±0.053 |
> | PMM WMod | P | P | 0.522±0.055 | 0.607±0.053 | 0.847±0.050 | 0.877±0.039 |
> | **Multimodal** | | | | | | |
> | SimCLR | P+A | P+A | 0.445±0.048 | 0.596±0.053 | 0.721±0.056 | 0.753±0.051 |
> | PMM WMod | P+A | P+A | 0.486±0.054 | 0.562±0.055 | 0.753±0.056 | 0.795±0.047 |
> | [NEW]  Abbaspourazad et al.| P+A | P+A | 0.417±0.044 | 0.551±0.056 | 0.668±0.062 | 0.753±0.050 |
> | [NEW]  RelCon | P+A | P+A | 0.573±0.052 | 0.640±0.051 | **0.910±0.035** | **0.918±0.031** |
> | CLIP | P\|A | P+A | 0.496±0.048 | 0.618±0.053 | **0.910±0.034** | **0.918±0.031** |
> | COCOA | P\|A | P+A | 0.508±0.050 | 0.607±0.053 | 0.778±0.053 | 0.808±0.046 |
> | CroSSL | P\|A | P+A | 0.430±0.045 | 0.494±0.052 | 0.783±0.053 | 0.808±0.047 |
> | FOCAL | P\|A | P+A | _0.596±0.049_ | _0.708±0.049_ | 0.888±0.041 | 0.904±0.035 |
> | SLIP | P\|A | P+A | 0.500±0.039 | 0.652±0.051 | 0.848±0.046 | 0.863±0.040 |
> | **ProtoMM** | P\|A | P+A | **0.622±0.050** | **0.719±0.048** | **0.910±0.035** | **0.918±0.031** |

---

> ### Author Response · Authors · 2025-11-26
>
> # 1. [Continued] Baselines
> ### Addressing:
> * W1a: “The main weakness of the work is the fact that the baselines are outdated. There exist several modern works that use foundation models for biosignals; in all cases the procedure is different to the one proposed in this work, but we cannot understand how well the proposed method works without modern baselines.”
> * W1b: “There exist several more recent SSL methods operating on biosignals that could be explored or adapted”
>
> ### Response:
> [Continued]
>
> **Table 3: Results on PPG-DaLiA Dataset**
>
> | Model | In | Out | Activity (9) F1 | Activity (9) Acc | HR MAE (↓) | HR R2 (↑) |
> | :--- | :---: | :---: | :---: | :---: | :---: | :---: |
> | **Unimodal** | | | | | | |
> | SimCLR | A | A | 0.559±0.005 | 0.560±0.004 | 15.37±0.10 | 0.101±0.015 |
> | SimCLR | P | P | 0.302±0.004 | 0.399±0.004 | **8.14±0.07** | **0.670±0.008** |
> | PMM WMod | A | A | 0.586±0.005 | 0.577±0.004 | 15.45±0.10 | 0.097±0.015 |
> | PMM WMod | P | P | 0.285±0.004 | 0.397±0.004 | _8.29±0.07_ | **0.670±0.007** |
> | **Multimodal** | | | | | | |
> | SimCLR | P+A | P+A | 0.542±0.005 | 0.536±0.004 | 9.91±0.08 | 0.534±0.009 |
> | PMM WMod | P+A | P+A | 0.543±0.005 | 0.547±0.005 | 15.07±0.09 | 0.155±0.014 |
> | [NEW]  Abbaspourazad et al. | P+A | P+A | 0.444±0.005 | 0.453±0.004 | 15.38±0.10 | 0.103±0.014 |
> | [NEW]  RelCon | P+A | P+A | 0.433±0.005 | 0.461±0.005 | 9.63±0.08 | 0.578±0.009 |
> | CLIP | P\|A | P+A | _0.640±0.005_ | _0.624±0.004_ | 9.37±0.07 | 0.609±0.008 |
> | COCOA | P\|A | P+A | 0.620±0.005 | 0.602±0.004 | 9.43±0.07 | 0.615±0.008 |
> | CroSSL | P\|A | P+A | 0.553±0.005 | 0.540±0.004 | 11.47±0.08 | 0.464±0.010 |
> | FOCAL | P\|A | P+A | 0.607±0.005 | 0.606±0.004 | 11.66±0.08 | 0.449±0.010 |
> | SLIP | P\|A | P+A | 0.633±0.004 | 0.625±0.004 | 8.89±0.08 | 0.634±0.008 |
> | **ProtoMM** | P\|A | P+A | **0.656±0.004** | **0.638±0.004** | 8.74±0.07 | 0.648±0.008 |
>
> *Key: A=Accel, P=PPG; + designates concatenation, \| designates separate encoders.*
>
>
> We would like to emphasize that many of our baselines are recent methods for multimodal time-series. FOCAL, CroSSL, COCOA are multimodal time-series SSL methods published at NeurIPS 2023, WSD2024, UbiComp 2022, respectively. Many of the methods suggested by the reviewer are not applicable. For the SSL approaches: [5] is a RAG approach, which requires paired text data, which we do not have as our source data were collected in free living conditions; [6] requires an “anchor” modality which gives ground truth context (i.e. video), which we do not have; [7] is a fully supervised method, and it is not obvious how to extend it to SSL. For the foundation models, [3] uses minutely data, rather than raw sensor used in our setting, [4] is an application of RelCon, which we now include.

---

> ### Author Response · Authors · 2025-11-26
>
> # 2. Additional datasets with other modalities
> ### Addressing:
> * W2: “The paper has evaluated only two modalities, which makes claims such as “general multimodal framework” not convincing.”
> * W3: “The datasets seem outdated and it seems that there exist newer and more within scope datasets.”
>
> ### Response:
> Thank you for giving us the opportunity to clarify. In our setting, we are focused on learning a transferable representation for multiple time-series modalities. As such, we pre-train on a large-scale PPG + Accelerometry dataset and evaluate our method across other datasets with PPG + Accelerometry, such as WESAD and PPG-DaLiA. We note that EEVR [8] and Stressid [9] do not contain PPG + Accelerometry time-series data.
>
> However, we do agree that adding further datasets with different modality combinations would be compelling. During the rebuttal period, we have pre-trained on a new dataset, the Moving Object Detection (MOD) dataset [17]. It provides seismic vibration and audio recordings generated by different types of moving vehicles. We utilize vehicle classification as the downstream task, and consider multiple evaluation protocols including linear probing, kNN classification and clustering.
>
> Our results remain consistent, and ProtoMM demonstrates itself as the best multi-modal method. We add the table below for easy reference.
>
>
> **Table: Performance evaluation of models pretrained on MOD dataset across different evaluation protocols (Linear-probe, K-Nearest Neighbors, and Clusterability).**
>
> | Method | Metric | CLIP | COCOA | CroSSL | FOCAL | SLIP | ProtoMM |
> | :--- | :---: | :---: | :---: | :---: | :---: | :---: | :---: |
> | **Linear-probe** | F1 (↑) | 0.702±0.012 | 0.220±0.008 | 0.702±0.012 | 0.639±0.013 | _0.737±0.011_ | **0.782±0.011** |
> | | Acc (↑) | 0.719±0.011 | 0.306±0.012 | 0.719±0.011 | 0.652±0.012 | _0.749±0.011_ | **0.787±0.011** |
> | **KNN** | F1 (↑) | 0.315±0.011 | 0.265±0.012 | 0.262±0.012 | 0.318±0.012 | _0.533±0.013_ | **0.564±0.012** |
> | | Acc (↑) | 0.334±0.012 | 0.288±0.012 | 0.286±0.012 | 0.330±0.012 | _0.517±0.013_ | **0.574±0.012** |
> | **Clusterability** | ARI (↑) | 0.086±0.009 | 0.093±0.009 | 0.070±0.008 | 0.035±0.013 | **0.190±0.018** | _0.188±0.032_ |
> | | NMI (↑) | 0.158±0.012 | 0.158±0.010 | 0.140±0.011 | 0.052±0.016 | _0.285±0.017_ | **0.298±0.033** |
>
> *Note: **Bold** indicates best performance; _Italics_ indicates second best.*

---

> ### Author Response · Authors · 2025-11-26
>
> # 3. Further justification on prototypes and sinkhorn
> ### Addressing:
> * W4: “While intuitively well-motivated, the paper lacks a deeper theoretical justification of why prototype consistency across modalities results in better disentanglement or regularization...”
> * W7: “Although there exist some visualizations, we cannot understand the importance of training with both losses neither the importance of prototypes… For example, we cannot understand if the shared prototype space is truly modality-agnostic, or if some modality-specific subclusters have emerged.”
> * W8: “Prototypes are the main contribution of the work but we cannot understand how they operate. Besides visualization, we would now what happens for instance when we have one dominant modality or how many we need to have or if we need additional latents to represent them or what would happen if we have longer sequences.”
>
> ### Response:
>
> Thank you for the  great discussion points.
> (1) **We present new visualization in Figure 4, demonstrating that our shared prototype space disentangles information: most prototypes focus on modality-agnostic information, while some will capture modality-specific features.** This is beneficial because the prototypes mostly capture the shared underlying generative state, but still retains information that may be more relevant to a given modality (i.e. quasiperiodicity is relevant for PPG but not Accel).
>
> (2) **Next, we have now added a derivation of our Multimodal Prototype Prediction Loss from log-likelihood** into Appendix Section A.4, which demonstrates our objective formally maximizes the ELBO of the joint data distribution while the prototypes act as an information bottleneck to filter out semantically-irrelevant noise. This derivation can be found below.
>
> Our objective is to find the network parameters $\theta$ that maximizes the log-likelihood function of the observed $i$ samples, composed of $M$ modalities. :
> $$
> \theta^* = \arg\max_\theta \sum_{i} \sum_{a=1}^A \sum_{m=1}^M
> \log p(\mathbf{X}^{(a)}_{i,m} ;\theta)
> $$
>
>
> We assume that the observed data modality $\mathbf{X}^{(a)}\_{i,m}$
> is related to some combination of modality-specific and modality-shared prototypes
> $\mathbf{P} = {\mathbf{P}\_m \cup \mathbf{P}\_s}$ where
> $\mathbf{p}^{(a)}\_{i,m}$ is the optimal prototype assignment of
> $\mathbf{X}^{(a)}\_{i,m}$. We abbreviate $\mathbf{p}^{(a)}\_{i,m}$
> to $\mathbf{p}$ for brevity.
>
> $$
> = \arg\max_\theta
> \sum_{i} \sum\_{a=1}^A \sum\_{m=1}^M
> \log \sum\_{\mathbf{p} \in \{\mathbf{P}\_m \cup \mathbf{P}\_s\}}
> p(\mathbf{X}^{(a)}\_{i,m}, \mathbf{p} ;\theta)
> $$
>
> Given a $Q(\mathbf{p})$ probability simplex for $\mathbf{p}$,
> such that $\sum\_{\mathbf{p} \in \mathbf{P}} Q(\mathbf{p}) = 1$,
>
> $$
> = \arg\max_\theta
> \sum_{i} \sum\_{a=1}^A \sum\_{m=1}^M
> \log \sum\_{\mathbf{p} \in \{\mathbf{P}\_m \cup \mathbf{P}\_s\}}
> Q(\mathbf{p})
> \frac{ p(\mathbf{X}^{(a)}\_{i,m}, \mathbf{p} ;\theta)}{Q(\mathbf{p})}
> $$
>
> With Jensen's inequality,
>
> $$
> \ge \arg\max_\theta
> \sum\_{i} \sum\_{a=1}^A \sum_{m=1}^M
> \sum\_{\mathbf{p} \in \{\mathbf{P}\_m \cup \mathbf{P}\_s\}}
> Q(\mathbf{p})
> \log \frac{ p(\mathbf{X}^{(a)}\_{i,m}, \mathbf{p} ;\theta)}{Q(\mathbf{p})}
> $$
>
> Therefore, removing constants, we should maximize:
> $$
> \sum\_{i} \sum\_{m=1}^M \sum\_{\textbf{p} \in \{\textbf{P}\_m \cup \textbf{P}\_{s}\}}  Q(\textbf{p}) \log  { p(\textbf{X}^{(a)}\_{i,m} , \textbf{p} ;\theta)}
> $$
>
> $$
> = \sum\_{i} \sum\_{m=1}^M \left( \sum\_{\textbf{p} \in \textbf{P}\_s}  Q(\textbf{p}) \log  { p(\textbf{X}^{(a)}\_{i,m} , \textbf{p} ;\theta)} + \sum\_{\textbf{p} \in \textbf{P}\_m}  Q(\textbf{p}) \log  { p(\textbf{X}^{(a)}\_{i,m} , \textbf{p} ;\theta)} \right)
> $$
> Now with  $\mathbf{U}\_{m}^{(a)}= \mathrm{Softmax}({\mathbf{E}}_{m}^{(a)} \mathbf{P} / \tau)$, the softmax forms a probability distribution over $\mathbf{X}^{(a)}\_{i,m}$ and $\textbf{p}$, so we can substitute it in.
> $$
> = \sum\_{i} \sum\_{m=1}^M \sum\_{a=1}^A \left( \sum\_{\textbf{p} \in \textbf{P}\_s}  Q(\textbf{p}) \log  { \textbf{U}^{(a)}\_m} + \sum\_{\textbf{p} \in \textbf{P}\_m}  Q(\textbf{p}) \log  {\textbf{U}^{(a)}\_m} \right)
> $$
>
> Now with $\mathbf{V}_{m}^{(a)} = \mathrm{Sinkhorn}({\mathbf{E}}\_{m}^{(a)} \mathbf{P})$, the sinkhorn give us normalized soft prototype label assignments, we can substitute them in as being proportional to a true probability simplex.
>
> $$
> \propto \sum\_{i} \sum\_{m,n=1}^M \sum\_{a,b=1}^A \left( \sum\_{n \neq m} \textbf{V}^{(a)}\_n \log \textbf{U}^{(a)}\_m + \sum\_{b \neq a} \textbf{V}^{(b)}\_m \log \textbf{U}^{(a)}\_m \right) = - \mathcal{L}\_{\text{MPP}} * \beta
> $$
>
> This is now equivalent our negative loss function multiplied by a constant $\beta$. Therefore, by minimizing our $\mathcal{L}\_{\text{MPP}}$, we are maximizing this lower bound of the log-likelihood.

---

> ### Author Response · Authors · 2025-11-26
>
> # 3. [Continued] Further justification on prototypes and sinkhorn
> ### Addressing:
> * W4: “While intuitively well-motivated, the paper lacks a deeper theoretical justification of why prototype consistency across modalities results in better disentanglement or regularization...”
> * W7: “Although there exist some visualizations, we cannot understand the importance of training with both losses neither the importance of prototypes… For example, we cannot understand if the shared prototype space is truly modality-agnostic, or if some modality-specific subclusters have emerged.”
> * W8: “Prototypes are the main contribution of the work but we cannot understand how they operate. Besides visualization, we would now what happens for instance when we have one dominant modality or how many we need to have or if we need additional latents to represent them or what would happen if we have longer sequences.”
>
> ### Response:
> [Continued]
> **(3) Finally, we evaluate the class-purity of each prototype.**
>  For every prototype​, we retrieve its K nearest embeddings (from both PPG and ACC) and check whether these neighbors share the same class label.  This purity score quantifies whether prototypes form coherent semantic clusters.
> Across datasets, we observe high purity scores, showing that prototypes consistently group semantically similar physiological segments.
> **Table: Prototype class purity (%)  over binary classification tasks.**
>
> | Dataset | Task | Mod | Top-3 | Top-5 | Top-10 |
> | :--- | :--- | :---: | :---: | :---: | :---: |
> | **MOODS** | Stress (2) | A | 41.95 | 23.92 | 9.21 |
> | | | P | 36.56 | 16.33 | 4.56 |
> | | Activity (2) | A | 83.57 | 77.65 | 70.13 |
> | | | P | 79.80 | 71.16 | 61.15 |
> | **WESAD** | Stress (2) | A | 53.74 | 33.12 | 13.26 |
> | | | P | 64.85 | 45.62 | 27.38 |
>
> *Note: Mod = Modality ; (A)ccel, (P)PG*
>
> **(4) Our final conclusion is that the theoretical justification lies in how learning prototype consistency discretizes the embedding space.** Without the prototypes, ProtoMM reverts to SLIP, relying on a continuous embedding space optimized via standard contrastive objectives (e.g., SimCLR/CLIP). However, recent literature highlights that such contrastive learning can often fail to learn an overlapping multimodal embedding due to the “modality gap” [11] (a persistent separation in the embedding caused by the contrastive objective encouraging the retention of modality-specific noise) and “class collision" [12, 13] (contrastive learning pushes semantically similar instances because they share a batch and are in the negative set).
> ProtoMM unifies these insights to project embeddings onto a finite set of shared prototypes. This acts as a discrete bottleneck that filters out modality-specific semantically-irrelevant noise [14], bridging the modality gap. Simultaneously, it clusters semantically similar instances, resolving class collision. This discretization is particularly well-suited for physiological time-series, which are composed of recurring motifs (e.g., neuroactivity patterns [15] or gait cycles [16]) that are naturally encoded by a discrete dictionary.

---

> ### Author Response · Authors · 2025-11-26
>
> # 4. Additional Ablations
> ### Addressing:
> * W4: Furthermore, there is no discussion on the convergence or stability of the Sinkhorn-based assignment
> * W5: “Several ablations are missing (hyperparam robustness, number of prototypes, temperature, choice of \alpha, etc). These could really help understand what works and why in the proposed method…It remains unclear whether ProtoMM embeddings are better than contrastive baselines so a baseline with contrastive learning would help.”
>
> ### Response:
> Thank you for this discussion point. **In general, we did not want to rely on an extensive hyperparameter tuning in order to achieve strong results, so we chose default parameters while training.** This includes setting sinkhorn_iter=3 and temperature=0.1 to match prior work [18] and $\alpha=0.5$ as a simple way to combine the within and between-modality losses.
>
> However, we agree that a finer-grained hyperparameter analysis provides valuable information on its robustness, and during the rebuttal, we have run extensive hyperparameter ablation experiments.
>
> **We have updated our ablation study (Table 2) in the paper [Lines 378-391] to include further ablations on α=[0,0.25,0.5,0.67,0.75,1].** Here we see that ProtoMM achieves the best performance at α=0.5 and 0.67, showing that the unified MPP objective successfully captures both within- and between-modality information via a simple balanced combination of the two.
>
> | Dataset | Task | Metric | α=0 | α=0.25 | α=0.5 (ProtoMM) | α=0.67 | α=0.75 | α=1 |
> | :--- | :--- | :---: | :---: | :---: | :---: | :---: | :---: | :---: |
> | **MOODS** | Stress (2) | F1 (↑) | 0.461±0.005 | 0.463±0.005 | _0.532±0.006_ | 0.522±0.006 | **0.537±0.006** | 0.497±0.006 |
> | | | Acc (↑) | _0.604±0.006_ | **0.608±0.005** | 0.591±0.006 | 0.586±0.006 | 0.600±0.006 | 0.591±0.006 |
> | | Activity (2) | F1 (↑) | 0.719±0.005 | 0.724±0.005 | **0.872±0.003** | 0.869±0.003 | _0.871±0.003_ | 0.855±0.004 |
> | | | Acc (↑) | 0.874±0.002 | 0.876±0.002 | **0.932±0.002** | 0.929±0.002 | _0.931±0.002_ | 0.923±0.002 |
> | **WESAD** | Stress (4) | F1 (↑) | 0.578±0.057 | 0.589±0.058 | **0.622±0.050** | 0.577±0.054 | 0.526±0.047 | _0.612±0.055_ |
> | | | Acc (↑) | 0.663±0.050 | _0.685±0.052_ | **0.719±0.048** | 0.674±0.050 | 0.640±0.051 | _0.685±0.050_ |
> | | Stress (2) | F1 (↑) | 0.783±0.052 | 0.786±0.054 | _0.910±0.035_ | **0.938±0.031** | 0.800±0.051 | 0.858±0.049 |
> | | | Acc (↑) | 0.808±0.045 | 0.822±0.044 | _0.918±0.031_ | **0.945±0.027** | 0.822±0.045 | 0.890±0.037 |
> | **PPG-DaLiA**| Activity (9) | F1 (↑) | 0.496±0.005 | 0.485±0.005 | 0.656±0.004 | _0.657±0.004_ | **0.662±0.004** | 0.618±0.005 |
> | | | Acc (↑) | 0.502±0.004 | 0.493±0.004 | 0.638±0.004 | **0.651±0.004** | _0.648±0.004_ | 0.603±0.004 |
> | | HR (R) | MAE (↓) | 10.45±0.08 | 10.76±0.08 | _8.74±0.07_ | **8.67±0.07** | 9.08±0.08 | 9.14±0.08 |
> | | | R2 (↑) | 0.536±0.009 | 0.515±0.009 | _0.648±0.008_ | **0.654±0.007** | 0.618±0.008 | 0.611±0.008 |
>
> *Note: **Bold** indicates best performance; _Italics_ indicates second best.*

---

> ### Author Response · Authors · 2025-11-26
>
> # 4. [Continued] Additional Ablations
> ### Response:
> [Continued]
> Similarly, we investigate how the number of prototypes influences representation quality (Table 6 in Appendix). **We expand ablation study (Table 6 in Appendix) with number of prototypes P = [0.3k, 1k, 3k, 10k, 30k, 100k]** Across datasets, we observe stable performance when P is greater or equal to 1k. The performance gains for extremely large sets of prototypes (30k, 100k) are small (e.g. 1.1% improvement on MOODS stress F1, 1.2% improvement on WESAD binary stress F1). This confirms that ProtoMM does not rely on large prototype sets to function well. Overall, this demonstrates that ProtoMM is highly robust to prototype-set size.
>
> **Table: Effect of the number of prototypes (P) on downstream performance.**
>
> | Dataset | Task | Metric | P=0.3k | P=1k | P=3k | P=10k (ProtoMM) | P=30k | P=100k |
> | :--- | :--- | :---: | :---: | :---: | :---: | :---: | :---: | :---: |
> | **MOODS** | Stress (2) | F1 (↑) | 0.448±0.005 | _0.532±0.006_ | _0.532±0.006_ | _0.532±0.006_ | 0.529±0.006 | **0.543±0.006** |
> | | | Acc (↑) | **0.616±0.005** | 0.587±0.006 | 0.593±0.006 | 0.591±0.006 | 0.586±0.005 | _0.597±0.006_ |
> | | Activity (2) | F1 (↑) | 0.619±0.005 | _0.871±0.003_ | 0.868±0.003 | **0.872±0.003** | _0.871±0.003_ | 0.869±0.003 |
> | | | Acc (↑) | 0.855±0.002 | _0.931±0.002_ | 0.929±0.002 | **0.932±0.002** | _0.931±0.002_ | 0.930±0.002 |
> | **WESAD** | Stress (4) | F1 (↑) | 0.518±0.059 | **0.641±0.053** | 0.556±0.036 | _0.622±0.050_ | 0.546±0.049 | 0.607±0.053 |
> | | | Acc (↑) | 0.596±0.055 | **0.719±0.047** | _0.697±0.050_ | **0.719±0.048** | 0.618±0.052 | _0.697±0.049_ |
> | | Stress (2) | F1 (↑) | 0.712±0.059 | _0.910±0.035_ | 0.905±0.037 | _0.910±0.035_ | **0.922±0.034** | 0.824±0.048 |
> | | | Acc (↑) | 0.767±0.049 | _0.918±0.032_ | _0.918±0.032_ | _0.918±0.031_ | **0.932±0.030** | 0.836±0.044 |
> | **PPG-DaLiA**| Activity (9) | F1 (↑) | 0.498±0.005 | **0.661±0.004** | _0.656±0.004_ | _0.656±0.004_ | 0.645±0.004 | **0.661±0.004** |
> | | | Acc (↑) | 0.503±0.004 | **0.645±0.004** | 0.634±0.004 | 0.638±0.004 | 0.626±0.004 | _0.641±0.004_ |
> | | HR (R) | MAE (↓) | 10.58±0.08 | **8.59±0.07** | 8.84±0.07 | _8.74±0.07_ | 8.82±0.07 | 8.78±0.07 |
> | | | R2 (↑) | 0.529±0.009 | **0.662±0.007** | 0.641±0.008 | _0.648±0.008_ | 0.638±0.008 | 0.643±0.008 |
>
> *Note: **Bold** indicates best performance; _Italics_ indicates second best.*
>
> **We examine the impact of Sinkhorn-Knopp normalization with two ablations (Table 7 in Appendix)**: replacing Sinkhorn with softmax assignment, and disenabling Sinkhorn entirely by setting number of iterations as 0. Both modifications yield large performance drops across datasets (e.g. PPG-DaLiA Activity F1 drops by 30.1% and 26.9%, respectively). This highlights that Sinkhorn-Knopp is a critical mechanism for avoiding collapse and ensuring stable alignment.
>
> **Table: Ablation study on Sinkhorn-Knopp normalization (MOODS)**
>
> | Setting | Stress (2) F1 | Stress (2) Acc | Activity (2) F1 | Activity (2) Acc |
> | :--- | :---: | :---: | :---: | :---: |
> | **w/ Sinkhorn (ProtoMM)** | **0.532±0.006** | **0.591±0.006** | **0.872±0.003** | **0.932±0.002** |
> | Sinkhorn → Softmax | -11.0% | +2.6% | -24.2% | -7.8% |
> | w/o Sinkhorn | -11.6% | +1.1% | -11.9% | -4.5% |
>
> **Table: Ablation study on Sinkhorn-Knopp normalization (WESAD)**
>
> | Setting | Stress (4) F1 | Stress (4) Acc | Stress (2) F1 | Stress (2) Acc |
> | :--- | :---: | :---: | :---: | :---: |
> | **w/ Sinkhorn (ProtoMM)** | **0.622±0.050** | **0.719±0.048** | **0.910±0.035** | **0.918±0.031** |
> | Sinkhorn → Softmax | -9.2% | -7.8% | -14.6% | -11.9% |
> | w/o Sinkhorn | -17.0% | -15.6% | -21.8% | -16.4% |
>
> **Table: Ablation study on Sinkhorn-Knopp normalization (PPG-DaLiA)**
>
> | Setting | Activity (9) F1 | Activity (9) Acc | HR MAE (↓) | HR R2 (↑) |
> | :--- | :---: | :---: | :---: | :---: |
> | **w/ Sinkhorn (ProtoMM)** | **0.656±0.004** | **0.638±0.004** | **8.74±0.07** | **0.648±0.008** |
> | Sinkhorn → Softmax | -30.1% | -26.3% | +28.0% | -24.9% |
> | w/o Sinkhorn | -26.9% | -22.4% | +25.6% | -23.6% |

---

> ### Author Response · Authors · 2025-11-26
>
> # 5. Fine-tuning Experiments
> ### Addressing:
> * W6: “All downstream evaluations use frozen encoders with linear probes thus making the method dependent on the quality of the encoder.”
> * Q2: “How does ProtoMM perform when fine-tuned end-to-end on downstream tasks, compared to linear probing?”
>
> ### Response:
> We thank the reviewer for this important discussion point. Our choice to prioritize linear probing is intentional. **All encoder architectures are constant across methods, and linear probe evaluation aligns with the standard evaluation protocols for foundation models [1, 2, 3, 4].** The primary goal of self-supervised learning is to learn a generalizable feature space that captures semantic structure before task-specific supervision. Linear probing effectively isolates the quality of this pre-trained representation. In contrast, full fine-tuning introduces significant stochasticity; it allows the encoder to essentially "re-learn" features from scratch, often masking the benefits of the pre-training or leading to overfitting on smaller downstream datasets.
> For completeness, we conducted a new comprehensive fine-tuning experiment during the rebuttal phase (see table below). The results support our hypothesis:
> 1. **Overfitting in Fine-Tuning**: Across almost all methods, fine-tuning yields performance that is either comparable to or worse than linear probing. This suggests that allowing the encoder to update destroys the robust, generalizable structure learned during pre-training, leading to overfitting on the specific downstream distribution.
> 2. **Unclear Evaluation**: None of the methods particularly demonstrate stand-out performance, demonstrating that fine-tuning is too stochastic to evaluate across methodologies.

---

> ### Author Response · Authors · 2025-11-26
>
> # 5. [Continued] Fine-tuning Experiments
> ### Addressing:
> * W6: “All downstream evaluations use frozen encoders with linear probes thus making the method dependent on the quality of the encoder.”
> * Q2: “How does ProtoMM perform when fine-tuned end-to-end on downstream tasks, compared to linear probing?”
>
> ### Response:
> [Continued]
>
> **Table 1: Finetuning Results on MOODS**
>
> | Model | In | Out | Stress (2) F1 | Stress (2) Acc | Activity (2) F1 | Activity (2) Acc |
> | :--- | :---: | :---: | :---: | :---: | :---: | :---: |
> | **Unimodal** | | | | | | |
> | SimCLR | A | A | 0.388±0.002 | **0.634±0.005** | 0.859±0.003 | 0.925±0.002 |
> | SimCLR | P | P | 0.388±0.002 | **0.634±0.005** | 0.737±0.005 | 0.879±0.002 |
> | PMM WMod | A | A | **0.404±0.003** | 0.626±0.005 | 0.844±0.004 | 0.922±0.002 |
> | PMM WMod | P | P | _0.390±0.002_ | **0.634±0.005** | 0.712±0.005 | 0.873±0.002 |
> | **Multimodal** | | | | | | |
> | SimCLR | P+A | P+A | 0.388±0.002 | **0.634±0.005** | _0.861±0.004_ | 0.925±0.002 |
> | PMM WMod | P+A | P+A | **0.404±0.003** | 0.627±0.005 | 0.859±0.004 | 0.926±0.002 |
> | [NEW] Abbaspourazad | P+A | P+A | 0.388±0.002 | **0.634±0.005** | 0.850±0.004 | 0.925±0.002 |
> | [NEW] RelCon | P+A | P+A | 0.388±0.002 | **0.634±0.005** | 0.860±0.003 | _0.927±0.002_ |
> | CLIP | P\|A | P+A | 0.388±0.002 | **0.634±0.005** | **0.867±0.003** | **0.929±0.002** |
> | COCOA | P\|A | P+A | 0.388±0.002 | **0.634±0.005** | 0.859±0.003 | 0.926±0.002 |
> | CroSSL | P\|A | P+A | 0.388±0.002 | **0.634±0.005** | 0.843±0.004 | 0.922±0.002 |
> | FOCAL | P\|A | P+A | 0.388±0.002 | **0.634±0.005** | 0.849±0.004 | 0.922±0.002 |
> | SLIP | P\|A | P+A | _0.390±0.002_ | **0.634±0.005** | 0.852±0.004 | 0.926±0.002 |
> | **ProtoMM** | P\|A | P+A | 0.388±0.002 | **0.634±0.005** | 0.857±0.004 | 0.921±0.002 |
>
> **Table 2: Finetuning Results on WESAD**
>
> | Model | In | Out | Stress (4) F1 | Stress (4) Acc | Stress (2) F1 | Stress (2) Acc |
> | :--- | :---: | :---: | :---: | :---: | :---: | :---: |
> | **Unimodal** | | | | | | |
> | SimCLR | A | A | 0.468±0.052 | 0.528±0.054 | 0.681±0.064 | 0.767±0.050 |
> | SimCLR | P | P | 0.504±0.040 | 0.618±0.051 | 0.807±0.049 | 0.822±0.045 |
> | PMM WMod | A | A | 0.472±0.053 | 0.528±0.055 | 0.793±0.055 | 0.836±0.043 |
> | PMM WMod | P | P | 0.496±0.046 | 0.517±0.053 | _0.890±0.041_ | _0.904±0.035_ |
> | **Multimodal** | | | | | | |
> | SimCLR | P+A | P+A | **0.631±0.054** | **0.730±0.050** | 0.859±0.044 | 0.877±0.037 |
> | PMM WMod | P+A | P+A | 0.485±0.053 | 0.562±0.054 | 0.793±0.055 | 0.836±0.043 |
> | [NEW] Abbaspourazad | P+A | P+A | 0.253±0.042 | 0.427±0.055 | 0.568±0.063 | 0.685±0.055 |
> | [NEW] RelCon | P+A | P+A | 0.512±0.054 | 0.596±0.053 | 0.706±0.063 | 0.795±0.045 |
> | CLIP | P\|A | P+A | 0.518±0.052 | 0.618±0.053 | 0.815±0.046 | 0.822±0.045 |
> | COCOA | P\|A | P+A | _0.575±0.048_ | _0.652±0.051_ | **0.935±0.031** | **0.945±0.026** |
> | CroSSL | P\|A | P+A | 0.306±0.046 | 0.472±0.057 | 0.568±0.069 | 0.740±0.053 |
> | FOCAL | P\|A | P+A | 0.402±0.042 | 0.562±0.053 | 0.611±0.066 | 0.740±0.050 |
> | SLIP | P\|A | P+A | 0.522±0.039 | 0.640±0.052 | 0.862±0.046 | 0.890±0.035 |
> | **ProtoMM** | P\|A | P+A | 0.559±0.047 | _0.652±0.053_ | 0.888±0.041 | _0.904±0.034_ |
>
> **Table 3: Finetuning Results on PPG-DaLiA**
>
> | Model | In | Out | Activity (9) F1 | Activity (9) Acc | HR MAE (↓) | HR R2 (↑) |
> | :--- | :---: | :---: | :---: | :---: | :---: | :---: |
> | **Unimodal** | | | | | | |
> | SimCLR | A | A | 0.587±0.004 | 0.586±0.004 | 18.33±0.11 | -0.196±0.010 |
> | SimCLR | P | P | 0.332±0.005 | 0.409±0.004 | 17.92±0.11 | -0.175±0.009 |
> | PMM WMod | A | A | 0.609±0.005 | 0.578±0.004 | **17.54±0.11** | **-0.122±0.008** |
> | PMM WMod | P | P | 0.324±0.004 | 0.442±0.005 | 18.70±0.11 | -0.235±0.011 |
> | **Multimodal** | | | | | | |
> | SimCLR | P+A | P+A | 0.594±0.004 | 0.583±0.004 | 18.63±0.11 | -0.225±0.011 |
> | PMM WMod | P+A | P+A | _0.626±0.004_ | 0.582±0.004 | 18.77±0.11 | -0.241±0.012 |
> | [NEW] Abbaspourazad | P+A | P+A | 0.580±0.005 | 0.567±0.004 | 18.32±0.11 | -0.197±0.010 |
> | [NEW] RelCon | P+A | P+A | 0.625±0.005 | 0.592±0.004 | 18.00±0.11 | -0.165±0.010 |
> | CLIP | P\|A | P+A | 0.554±0.004 | 0.570±0.004 | _17.66±0.11_ | _0.143±0.008_ |
> | COCOA | P\|A | P+A | 0.540±0.005 | 0.565±0.005 | 18.45±0.11 | -0.210±0.011 |
> | CroSSL | P\|A | P+A | 0.618±0.004 | _0.603±0.004_ | 18.46±0.11 | -0.209±0.011 |
> | FOCAL | P\|A | P+A | 0.618±0.005 | 0.599±0.004 | 18.42±0.11 | -0.202±0.011 |
> | SLIP | P\|A | P+A | 0.534±0.005 | 0.509±0.004 | 19.28±0.11 | -0.282±0.013 |
> | **ProtoMM** | P\|A | P+A | **0.665±0.004** | **0.622±0.004** | 18.38±0.11 | -0.201±0.011 |
>
> *Key: Encoder (In)put, Final Embedding (Out)put; (A)ccel, (P)PG; + designates concatenation, \| designates separate encoders.*

---

> ### Author Response · Authors · 2025-11-26
>
> # 6. Clarification
> ### Addressing:
> * W9/Q1/Q3
>
> ### Response:
>
> * W9: “Failure cases are not discussed.”
> We have added the following discussion into our new Appendix Section A.1 Limitations.
>
> While ProtoMM effectively aligns multimodal biosignals, it relies on the assumption that physiological dynamics can be discretized into a finite set of shared prototypes. This design choice imposes a specific inductive bias that works well for quasi-periodic signals (e.g., cardiac cycles in PPG, gait in accelerometry) but may be suboptimal for highly stochastic or non-stationary signals lacking recurring motifs. Additionally, our shared prototype constraint assumes a symmetric semantic overlap between modalities. In scenarios where one modality contains significant unique information irrelevant to the other (e.g., motion artifacts in accelerometry unrelated to heart rate), the strict alignment objective may suppress these modality-specific features if the prototype size is not sufficiently large to accommodate them
>
>
>
> * Q1: “it would be interesting to have the computational overhead of the Sinkhorn-based prototype updates compared to contrastive methods”
>
> We thank the reviewer for raising this thoughtful point. We computed the exact computational overhead and summarized our findings here. First, the Sinkhorn step contributes only $\(O(BP)\)$ cost per iteration. With a fixed small number of iterations (n_iter = 3), its overhead is negligible in practice. The dominant difference between CLIP-style contrastive learning and our prototype-based formulation comes from the matrix multiplication terms:
> $$
> \text{CLIP: } O(E B^2) \quad\text{vs.}\quad \text{Sinkhorn-based Prototype Objective } O(E B P).
> $$
>
> However, our ablation on the number of prototypes (Table 6 in Appendix) shows that **ProtoMM does not require large prototype sets**. We sweep ${P = [0.3\text{k}, 1\text{k}, 3\text{k}, 10\text{k}, 30\text{k}, 100\text{k}]}$ and find that performance stabilizes as soon as ${P \ge 1\text{k}}$. Increasing $P$ to 30k–100k offers only marginal gains. Therefore, while large prototype sets naturally incur higher cost, as the ${O(BEP)}$ term, **ProtoMM operates effectively and efficiently with moderate prototype sizes (1k–10k)**. These settings provide strong and stable performance while keeping the actual computational overhead modest.
>
> We provide a detailed proof below. Let batch size be B, embedding dimension E, and number of prototypes P. Embeddings E have shape [B,E] and the set of prototypes P has shape [E,P].
>
>  **1. Prototype-Based Objective**
>
> **1.1 Prototype scores**
>
> Dense matrix multiplication:
> $$
> S = EP \in \mathbb{R}^{B \times P}, \quad E \in \mathbb{R}^{B \times E},\ P \in \mathbb{R}^{E \times P}
> $$
> Cost:
> $
> \mathcal{O}(BEP)
> $
>
>  **1.2 Softmax / log-softmax**
>
> Row-wise softmax over $[B \times P]$:
> $
> \mathcal{O}(BP)
> $
>
> **1.3 Sinkhorn normalization**
> Each iteration normalizes all ${B \times P}$ entries:
> $
> \mathcal{O}(BP)
> $
> With a small constant number of iterations ${T}$ (e.g., ${T=3}$), total is still
> $
> \mathcal{O}(BP)
> $
>
> **1.4 Cross-entropy between assignments**
> Elementwise multiply and sum across ${B \times P}$ elements:
> $
> \mathcal{O}(BP)
> $
>
>  **Overall complexity**
>
> $
> \boxed{\mathcal{O}(BEP + BP + BP + BP) = \mathcal{O}(BEP)}
> $
> dominated by the embedding–prototype multiplication.
>
>  **2. CLIP-Style Contrastive Learning**
>
> **2.1 Similarity logits**
> $$
> L = XY^\top \in \mathbb{R}^{B \times B}
> $$
> Cost:
> $
> \mathcal{O}(B^2 E)
> $
>
> **2.2 Cross-entropy with labels**
> Cross-entropy over ${B \times B}$ elements:
> $
> \mathcal{O}(B^2)
> $
>
>  **Overall complexity**
>
> $
> \boxed{\mathcal{O}(B^2 E + B^2) = \mathcal{O}(B^2 E)}
> $
> dominated by the quadratic similarity computation.
>
> **3. Summary**
>
> - **Prototype-based loss (ours):**
> $$
>   \mathcal{O}(BEP)
> $$
>   Linear in $B$. Sinkhorn adds only ${O(BP)}$, negligible relative to ${BEP}$.
>
> - **CLIP-style contrastive loss:**
> $$
>   \mathcal{O}(B^2 E)
> $$
>   Quadratic in $B$ due to full pairwise similarities.

---

> ### Author Response · Authors · 2025-11-26
>
> # 6. [Continued] Clarification
> ### Addressing:
> * W9/Q1/Q3
>
> ### Response:
> [Continued]
> * Q3: “Are the learned prototypes stable across training runs, or do they vary significantly with initialization?”
>
> We have found the learned prototypes to be stable across training runs. In Figure 5 in Appendix, We visualize the prototype space with t-SNE across 10 different random seeds (0–900).
> The scatter patterns remain nearly identical, confirming that the learned prototypes
> form a stable structure independent of the initialization. This is because the prototypes act as learned cluster centers, and similar to k-means, although the prototypes are randomly initialized, our prototypes will be stable after convergence.
>
>
>
> # References
> [1] Xu et. al., RelCon: Relative Contrastive Learning for a Motion Foundation Model for Wearable Data, ICLR 2025
> [2] Abbaspourazad, et. al., Large-scale Training of Foundation Models for Wearable Biosignals, ICLR 2024
> [3] Narayanswamy et. al., Scaling wearable foundation models, ICLR 2025
> [4] Saha et. al., Pulse-PPG: An Open-Source Field-Trained PPG Foundation Model for Wearable Applications across Lab and Field Settings, ACM on Interactive, Mobile, Wearable and Ubiquitous Technologies
> [5] Tag et. al., Electrocardiogram Report Generation and Question Answering via Retrieval-Augmented Self-Supervised Modeling, ICASSP 2025
> [6] Mordacq et. al., ADAPT: Multimodal Learning for Detecting Physiological Changes under Missing Modalities, MIDL 2024
> [7] Shen et. al., CIMSleepNet: Robust Sleep Staging over Incomplete Multimodal Physiological Signals via Contrastive Imagination, NeurIPS 2024
> [8] EEVR: A Dataset of Paired Physiological Signals and Textual Descriptions for Joint Emotion Representation Learning, NeurIPS 2024
> [9] Stressid: a multimodal dataset for stress identification, NeurIPS 2023
> [10] WildPPG (A Real-World PPG Dataset of Long Continuous Recordings), NeurIPS 2024
> [11] Liang, Victor Weixin, et al. "Mind the gap: Understanding the modality gap in multi-modal contrastive representation learning." Advances in Neural Information Processing Systems 35 (2022): 17612-17625.
> [12] Li, J., et al. "Prototypical Contrastive Learning of Unsupervised Representations." ICLR, 2021.
> [13] Wang, Yifei, et al. "Chaos is a ladder: A new theoretical understanding of contrastive learning via augmentation overlap." ICLR (2022).
> [14] Xia, Yan, et al. "Achieving cross modal generalization with multimodal unified representation." Advances in Neural Information Processing Systems 36 (2023): 63529-63541.
> [15] Pradeepkumar, Jathurshan, et al. "Tokenizing Single-Channel EEG with Time-Frequency Motif Learning." NeurIPS 2025 Workshop on Learning from Time Series for Health.
> [16] Haresamudram, Harish, Irfan Essa, and Thomas Ploetz. "Towards learning discrete representations via self-supervision for wearables-based human activity recognition." Sensors 24.4 (2024): 1238.
> [17] Liu, Shengzhong, et al. "Focal: Contrastive learning for multimodal time-series sensing signals in factorized orthogonal latent space." Advances in Neural Information Processing Systems 36 (2023): 47309-47338.
> [18] Caron, Mathilde, et al. "Unsupervised learning of visual features by contrasting cluster assignments." Advances in neural information processing systems 33 (2020): 9912-9924.

---

### Official Review · Reviewer_hWpB · 2025-11-07

**Soundness:** 2
**Presentation:** 3
**Contribution:** 3
**Rating:** 4
**Confidence:** 4

**Summary:**

This work introduces ProtoMM, a self-supervised multimodal model for time-series data that uses a shared prototype dictionary to align embeddings across modalities and capture both shared and modality-specific information. The model consistently outperforms unimodal and multimodal baselines in experiments across three datasets and downstream tasks, demonstrating strong interpretability and generalization.

**Strengths:**

1. The shared prototype dictionary provides an innovative alternative to contrastive learning methods, effectively addressing the limitations in negative pair construction in multimodal settings.

2. The model considers both shared and modality-specific information, deriving a shared cross-modal representation while preserving unique modality information.

**Weaknesses:**

1. Although the proposed method outperforms most baselines, the improvements are relatively small. Reporting standard deviations would help assess the statistical significance.

2. In Table 2, results are shown only for the alpha=0.5 setting and two extreme cases (0 and 1). Additional intermediate values should be included to verify whether the performance trend is consistent rather than random.

3. Same for Table 3. It would be great if the authors could show the results across a wider range of choices of alpha.

**Questions:**

Within-modality samples and cross-modality pairs are different. The within-modality samples are derived from augmentations, while each modality is from distinct sources. Therefore, the semantic meaning of distance (or similarity) differs.
Has the model design considered this discrepancy?

---

> ### Author Response · Authors · 2025-11-26
>
> # 1. Significant Performance Increase
> ### Addressing:
> * W1: “Although the proposed method outperforms most baselines, the improvements are relatively small. Reporting standard deviations would help assess the statistical significance.”
>
> ### Response:
> Thank you for the great suggestion. We have now performed additional analysis by bootstrapping random subsets of the test set 1000 times and computed the standard deviation of the resulting performance. All tables have now been updated with these values, demonstrating that although performance improvement may seem marginal, the differences are statistically significant.
>
> **Table 1: Results on MOODS Dataset**
>
> | Model | In | Out | Stress (2) F1 (↑) | Stress (2) Acc (↑) | Activity (2) F1 (↑) | Activity (2) Acc (↑) |
> | :--- | :---: | :---: | :---: | :---: | :---: | :---: |
> | **Unimodal** | | | | | | |
> | SimCLR | A | A | 0.477±0.006 | 0.598±0.006 | 0.861±0.004 | 0.925±0.002 |
> | SimCLR | P | P | _0.527±0.006_ | 0.607±0.006 | 0.655±0.005 | 0.857±0.002 |
> | PMM WMod | A | A | 0.464±0.006 | 0.604±0.006 | 0.857±0.004 | 0.924±0.002 |
> | PMM WMod | P | P | 0.469±0.005 | 0.611±0.005 | 0.595±0.005 | 0.848±0.003 |
> | **Multimodal** | | | | | | |
> | SimCLR | P+A | P+A | 0.488±0.006 | 0.601±0.006 | 0.849±0.004 | 0.919±0.002 |
> | PMM WMod | P+A | P+A | 0.467±0.006 | 0.600±0.006 | 0.836±0.004 | 0.913±0.002 |
> | [NEW] Abbaspourazad et al. | P+A | P+A | 0.445±0.005 | _0.614±0.005_ | 0.681±0.005 | 0.865±0.002 |
> | [NEW]  RelCon | P+A | P+A | 0.520±0.006 | 0.589±0.005 | 0.748±0.004 | 0.880±0.002 |
> | CLIP | P\|A | P+A | 0.524±0.006 | 0.578±0.006 | _0.869±0.003_ | _0.929±0.002_ |
> | COCOA | P\|A | P+A | 0.520±0.006 | 0.579±0.005 | 0.858±0.003 | 0.924±0.002 |
> | CroSSL | P\|A | P+A | 0.461±0.005 | **0.622±0.006** | 0.751±0.004 | 0.882±0.002 |
> | FOCAL | P\|A | P+A | 0.510±0.006 | 0.588±0.006 | 0.853±0.003 | 0.921±0.002 |
> | SLIP | P\|A | P+A | 0.524±0.006 | 0.586±0.006 | 0.867±0.003 | _0.929±0.002_ |
> | **ProtoMM** | P\|A | P+A | **0.532±0.006** | 0.591±0.006 | **0.872±0.003** | **0.932±0.002** |
>
> **Table 2: Results on WESAD Dataset**
>
> | Model | In | Out | Stress (4) F1 (↑) | Stress (4) Acc (↑) | Stress (2) F1 (↑) | Stress (2) Acc (↑) |
> | :--- | :---: | :---: | :---: | :---: | :---: | :---: |
> | **Unimodal** | | | | | | |
> | SimCLR | A | A | 0.557±0.053 | 0.629±0.052 | 0.793±0.055 | 0.836±0.044 |
> | SimCLR | P | P | 0.348±0.034 | 0.517±0.053 | 0.692±0.056 | 0.699±0.054 |
> | PMM WMod | A | A | 0.533±0.057 | 0.584±0.053 | 0.675±0.060 | 0.726±0.053 |
> | PMM WMod | P | P | 0.522±0.055 | 0.607±0.053 | 0.847±0.050 | 0.877±0.039 |
> | **Multimodal** | | | | | | |
> | SimCLR | P+A | P+A | 0.445±0.048 | 0.596±0.053 | 0.721±0.056 | 0.753±0.051 |
> | PMM WMod | P+A | P+A | 0.486±0.054 | 0.562±0.055 | 0.753±0.056 | 0.795±0.047 |
> | [NEW]  Abbaspourazad et al.| P+A | P+A | 0.417±0.044 | 0.551±0.056 | 0.668±0.062 | 0.753±0.050 |
> | [NEW]  RelCon | P+A | P+A | 0.573±0.052 | 0.640±0.051 | **0.910±0.035** | **0.918±0.031** |
> | CLIP | P\|A | P+A | 0.496±0.048 | 0.618±0.053 | **0.910±0.034** | **0.918±0.031** |
> | COCOA | P\|A | P+A | 0.508±0.050 | 0.607±0.053 | 0.778±0.053 | 0.808±0.046 |
> | CroSSL | P\|A | P+A | 0.430±0.045 | 0.494±0.052 | 0.783±0.053 | 0.808±0.047 |
> | FOCAL | P\|A | P+A | _0.596±0.049_ | _0.708±0.049_ | 0.888±0.041 | 0.904±0.035 |
> | SLIP | P\|A | P+A | 0.500±0.039 | 0.652±0.051 | 0.848±0.046 | 0.863±0.040 |
> | **ProtoMM** | P\|A | P+A | **0.622±0.050** | **0.719±0.048** | **0.910±0.035** | **0.918±0.031** |
>
> **Table 3: Results on PPG-DaLiA Dataset**
>
> | Model | In | Out | Activity (9) F1 | Activity (9) Acc | HR MAE (↓) | HR R2 (↑) |
> | :--- | :---: | :---: | :---: | :---: | :---: | :---: |
> | **Unimodal** | | | | | | |
> | SimCLR | A | A | 0.559±0.005 | 0.560±0.004 | 15.37±0.10 | 0.101±0.015 |
> | SimCLR | P | P | 0.302±0.004 | 0.399±0.004 | **8.14±0.07** | **0.670±0.008** |
> | PMM WMod | A | A | 0.586±0.005 | 0.577±0.004 | 15.45±0.10 | 0.097±0.015 |
> | PMM WMod | P | P | 0.285±0.004 | 0.397±0.004 | _8.29±0.07_ | **0.670±0.007** |
> | **Multimodal** | | | | | | |
> | SimCLR | P+A | P+A | 0.542±0.005 | 0.536±0.004 | 9.91±0.08 | 0.534±0.009 |
> | PMM WMod | P+A | P+A | 0.543±0.005 | 0.547±0.005 | 15.07±0.09 | 0.155±0.014 |
> | [NEW]  Abbaspourazad et al. | P+A | P+A | 0.444±0.005 | 0.453±0.004 | 15.38±0.10 | 0.103±0.014 |
> | [NEW]  RelCon | P+A | P+A | 0.433±0.005 | 0.461±0.005 | 9.63±0.08 | 0.578±0.009 |
> | CLIP | P\|A | P+A | _0.640±0.005_ | _0.624±0.004_ | 9.37±0.07 | 0.609±0.008 |
> | COCOA | P\|A | P+A | 0.620±0.005 | 0.602±0.004 | 9.43±0.07 | 0.615±0.008 |
> | CroSSL | P\|A | P+A | 0.553±0.005 | 0.540±0.004 | 11.47±0.08 | 0.464±0.010 |
> | FOCAL | P\|A | P+A | 0.607±0.005 | 0.606±0.004 | 11.66±0.08 | 0.449±0.010 |
> | SLIP | P\|A | P+A | 0.633±0.004 | 0.625±0.004 | 8.89±0.08 | 0.634±0.008 |
> | **ProtoMM** | P\|A | P+A | **0.656±0.004** | **0.638±0.004** | 8.74±0.07 | 0.648±0.008 |
>
> *Key: A=Accel, P=PPG; + designates concatenation, \| designates separate encoders.*

---

> ### Author Response · Authors · 2025-11-26
>
> # 2. Capturing Within and Between-Modality Information with α
> ### Addressing:
> * W2: “In Table 2, results are shown only for the alpha=0.5 setting and two extreme cases (0 and 1). Additional intermediate values should be included to verify whether the performance trend is consistent rather than random.”
> * W3: “Same for Table 3. It would be great if the authors could show the results across a wider range of choices of alpha.”
>
> ### Response:
> We are grateful for the suggestion to show results across a wider range of choices of alpha. We have now added results by varying alpha in the following range: {0, 0.25, 0.5, 0.67, 0.75, 1} in Table 2. We observe that performance is generally higher when alpha is not equal to 0 or 1. This indicates the necessity of both within- and across-modality losses. The highest performance is achieved when alpha=0.5 or 0.67, showing that balancing both losses is optimal.
>
> | Dataset | Task | Metric | α=0 | α=0.25 | α=0.5 (ProtoMM) | α=0.67 | α=0.75 | α=1 |
> | :--- | :--- | :---: | :---: | :---: | :---: | :---: | :---: | :---: |
> | **MOODS** | Stress (2) | F1 (↑) | 0.461±0.005 | 0.463±0.005 | _0.532±0.006_ | 0.522±0.006 | **0.537±0.006** | 0.497±0.006 |
> | | | Acc (↑) | _0.604±0.006_ | **0.608±0.005** | 0.591±0.006 | 0.586±0.006 | 0.600±0.006 | 0.591±0.006 |
> | | Activity (2) | F1 (↑) | 0.719±0.005 | 0.724±0.005 | **0.872±0.003** | 0.869±0.003 | _0.871±0.003_ | 0.855±0.004 |
> | | | Acc (↑) | 0.874±0.002 | 0.876±0.002 | **0.932±0.002** | 0.929±0.002 | _0.931±0.002_ | 0.923±0.002 |
> | **WESAD** | Stress (4) | F1 (↑) | 0.578±0.057 | 0.589±0.058 | **0.622±0.050** | 0.577±0.054 | 0.526±0.047 | _0.612±0.055_ |
> | | | Acc (↑) | 0.663±0.050 | _0.685±0.052_ | **0.719±0.048** | 0.674±0.050 | 0.640±0.051 | _0.685±0.050_ |
> | | Stress (2) | F1 (↑) | 0.783±0.052 | 0.786±0.054 | _0.910±0.035_ | **0.938±0.031** | 0.800±0.051 | 0.858±0.049 |
> | | | Acc (↑) | 0.808±0.045 | 0.822±0.044 | _0.918±0.031_ | **0.945±0.027** | 0.822±0.045 | 0.890±0.037 |
> | **PPG-DaLiA**| Activity (9) | F1 (↑) | 0.496±0.005 | 0.485±0.005 | 0.656±0.004 | _0.657±0.004_ | **0.662±0.004** | 0.618±0.005 |
> | | | Acc (↑) | 0.502±0.004 | 0.493±0.004 | 0.638±0.004 | **0.651±0.004** | _0.648±0.004_ | 0.603±0.004 |
> | | HR (R) | MAE (↓) | 10.45±0.08 | 10.76±0.08 | _8.74±0.07_ | **8.67±0.07** | 9.08±0.08 | 9.14±0.08 |
> | | | R2 (↑) | 0.536±0.009 | 0.515±0.009 | _0.648±0.008_ | **0.654±0.007** | 0.618±0.008 | 0.611±0.008 |
>
> *Note: **Bold** indicates best performance; _Italics_ indicates second best.*

---

> ### Author Response · Authors · 2025-11-26
>
> # 3. Differences between pairs
> ### Addressing:
> * Q1: “Within-modality samples and cross-modality pairs are different. The within-modality samples are derived from augmentations, while each modality is from distinct sources. Therefore, the semantic meaning of distance (or similarity) differs. Has the model design considered this discrepancy?”
>
> ### Response:
> This is a great discussion point, and one of the strengths of our approach. By using a shared prototype space across both between- and within-modality pairs, we encourage the model to learn the underlying latent generative space. This makes sense in our approach because whether it be via augmentations or via differing sensor modalities, these transformations should be semantic preserving, as they are all describing an individual’s current physiological state and condition.
>
> However, we recognize that learning these relationships presents different levels of difficulty. Within-modality alignment via augmentations is very well studied [1], whereas between-modality alignment must overcome the “modality gap phenomenon” [2] between disparate sensor distributions. To this end, we introduce a new experiment, ablating our $\alpha$ value, which controls how much the within- vs. between-modality objectives apply to the overall loss function.
>
> **Table: ProtoMM achieves the best performance at α=0.5 and 0.67, showing that the unified MPP objective successfully captures both within- and between-modality information.**
>
> | Dataset | Task | Metric | α=0 | α=0.25 | α=0.5 (ProtoMM) | α=0.67 | α=0.75 | α=1 |
> | :--- | :--- | :---: | :---: | :---: | :---: | :---: | :---: | :---: |
> | **MOODS** | Stress (2) | F1 (↑) | 0.461±0.005 | 0.463±0.005 | _0.532±0.006_ | 0.522±0.006 | **0.537±0.006** | 0.497±0.006 |
> | | | Acc (↑) | _0.604±0.006_ | **0.608±0.005** | 0.591±0.006 | 0.586±0.006 | 0.600±0.006 | 0.591±0.006 |
> | | Activity (2) | F1 (↑) | 0.719±0.005 | 0.724±0.005 | **0.872±0.003** | 0.869±0.003 | _0.871±0.003_ | 0.855±0.004 |
> | | | Acc (↑) | 0.874±0.002 | 0.876±0.002 | **0.932±0.002** | 0.929±0.002 | _0.931±0.002_ | 0.923±0.002 |
> | **WESAD** | Stress (4) | F1 (↑) | 0.578±0.057 | 0.589±0.058 | **0.622±0.050** | 0.577±0.054 | 0.526±0.047 | _0.612±0.055_ |
> | | | Acc (↑) | 0.663±0.050 | _0.685±0.052_ | **0.719±0.048** | 0.674±0.050 | 0.640±0.051 | _0.685±0.050_ |
> | | Stress (2) | F1 (↑) | 0.783±0.052 | 0.786±0.054 | _0.910±0.035_ | **0.938±0.031** | 0.800±0.051 | 0.858±0.049 |
> | | | Acc (↑) | 0.808±0.045 | 0.822±0.044 | _0.918±0.031_ | **0.945±0.027** | 0.822±0.045 | 0.890±0.037 |
> | **PPG-DaLiA**| Activity (9) | F1 (↑) | 0.496±0.005 | 0.485±0.005 | 0.656±0.004 | _0.657±0.004_ | **0.662±0.004** | 0.618±0.005 |
> | | | Acc (↑) | 0.502±0.004 | 0.493±0.004 | 0.638±0.004 | **0.651±0.004** | _0.648±0.004_ | 0.603±0.004 |
> | | HR (R) | MAE (↓) | 10.45±0.08 | 10.76±0.08 | _8.74±0.07_ | **8.67±0.07** | 9.08±0.08 | 9.14±0.08 |
> | | | R2 (↑) | 0.536±0.009 | 0.515±0.009 | _0.648±0.008_ | **0.654±0.007** | 0.618±0.008 | 0.611±0.008 |
>
> *Note: **Bold** indicates best performance; _Italics_ indicates second best.*
>
>
> # References
> [1] Chen, Ting, et al. "A simple framework for contrastive learning of visual representations." International conference on machine learning. PmLR, 2020.
> [2] Liang, Victor Weixin, et al. "Mind the gap: Understanding the modality gap in multi-modal contrastive representation learning." Advances in Neural Information Processing Systems 35 (2022): 17612-17625.

---

### Official Review · Reviewer_rgHi · 2025-11-08

**Soundness:** 3
**Presentation:** 3
**Contribution:** 3
**Rating:** 6
**Confidence:** 3

**Summary:**

The authors present ProtoMM, a prototype-based self-supervised framework for multimodal time-series learning, designed specifically for pulse motion foundation models using PPG (photoplethysmography) and accelerometry data. The key idea is to replace traditional contrastive alignment (which requires explicit positive/negative pair mining) with a shared prototype dictionary that serves as semantic anchors across modalities. Each signal modality is encoded separately, projected into a shared prototype space, and trained via a Multimodal Prototype Prediction Loss, combining within-modality and between-modality consistency objectives. Empirical results across three datasets and six downstream tasks show that ProtoMM often outperforms unimodal and multimodal baselines. The authors also provide qualitative evidence of interpretability, showing that learned prototypes correspond to meaningful physiological and behavioral states.

**Strengths:**

- The paper presents a novel adaptation of SwAV’s prototype mechanism to a multimodal, multi-view learning setting.
- The study is highly relevant, addressing a practical and timely challenge in wearable sensing and multimodal foundation modeling.
- The proposed objective is generalizable, featuring a tunable parameter $\alpha$ that effectively controls the trade-off between within- and between-modality learning, with strong empirical support.
- The experimental evaluation is comprehensive, spanning multiple datasets and benchmarks under a consistent architecture with fair and transparent comparisons.
- The learned prototypes shows correspondence with semantically meaningful physiological patterns that enhance understanding of model behavior.
- The paper is clearly written, well-structured, and effectively motivates the proposed approach.

**Weaknesses:**

- The theoretical grounding of the approach is limited, as the benefits of prototypes over contrastive losses are supported mainly by empirical evidence. A formal analysis of how prototypes mitigate false negatives or enhance cross-modal alignment would substantially strengthen the work.
- The mathematical and biological motivation for the proposed Multimodal Prototype Prediction Loss is underdeveloped, with only a brief intuition provided in lines 194-195. A more detailed justification would better support the design and relevance of this loss function.
- Several closely related baseline methods (e.g., SLIP, FOCAL) are introduced only in the experimental section, rather than in the related work discussion. Including them earlier would clarify the paper’s positioning and help readers assess the novelty of the proposed approach more fairly.
- The ablation study is somewhat shallow, as it explores only three α values (0, 0.5, and 1). Testing a finer range would yield more informative results regarding the model’s sensitivity and stability.
- The interpretability evaluation is purely qualitative, lacking quantitative metrics such as clustering purity or correlations with physiological labels. Moreover, interpretability is not compared against baseline models, making it unclear whether these insights are unique to the proposed method.
- Minor typos in lines 058, 073.

**Questions:**

- How were the hyperparameters, such as $\alpha$ and the encoder architecture parameters, selected? Were they tuned empirically or determined through heuristic choices?
- In equations (5)-(7), there appear to be $MA^2$ individual within-modality losses and $M^2A^2$ between-modality losses. Would this not still lead to an imbalance between the two loss types, particularly when $\alpha = 0.5$?
- Could the authors clarify the definitions of $\mathbf{z}_t$ and $\mathbf{q}_s$ in equation (4)?
- In Lines 361–364, why would the hypothesized advantages of the prototype-based approach not extend to unimodal settings, especially given that empirical results show contrastive methods outperforming the prototype-based variant?
- In the interpretability experiment, could the authors provide quantitative validation showing that the learned prototypes correspond to specific physiological states?

---

> ### Author Response · Authors · 2025-11-26
>
> # 1. Further investigation into prototypes
> ### Addressing:
> * W1: “The theoretical grounding of the approach is limited, as the benefits of prototypes over contrastive losses are supported mainly by empirical evidence. A formal analysis of how prototypes mitigate false negatives or enhance cross-modal alignment would substantially strengthen the work.”
> * W2: “The mathematical and biological motivation for the proposed Multimodal Prototype Prediction Loss is underdeveloped, with only a brief intuition provided in lines 194-195. A more detailed justification would better support the design and relevance of this loss function.”
>
> ### Response:
> Thank you for the feedback, and we have added further analysis of our method and loss function into the paper (Appendix Section A.6). Firstly, false negatives are a non-issue because our method is a positive-only approach.
>
> Next, we have now added a derivation of our Multimodal Prototype Prediction Loss from log-likelihood into our paper, which demonstrates our objective formally maximizes the ELBO of the joint data distribution while the prototypes act as an information bottleneck to filter out semantically-irrelevant noise. This derivation can be found below.
>
> Our objective is to find the network parameters $\theta$ that maximizes the log-likelihood function of the observed $i$ samples, composed of $M$ modalities. :
> $$
> \theta^* = \arg\max_\theta \sum_{i} \sum_{a=1}^A \sum_{m=1}^M
> \log p(\mathbf{X}^{(a)}_{i,m} ;\theta)
> $$
>
>
> We assume that the observed data modality $\mathbf{X}^{(a)}\_{i,m}$
> is related to some combination of modality-specific and modality-shared prototypes
> $\mathbf{P} = {\mathbf{P}\_m \cup \mathbf{P}\_s}$ where
> $\mathbf{p}^{(a)}\_{i,m}$ is the optimal prototype assignment of
> $\mathbf{X}^{(a)}\_{i,m}$. We abbreviate $\mathbf{p}^{(a)}\_{i,m}$
> to $\mathbf{p}$ for brevity.
>
> $$
> = \arg\max_\theta
> \sum_{i} \sum\_{a=1}^A \sum\_{m=1}^M
> \log \sum\_{\mathbf{p} \in \{\mathbf{P}\_m \cup \mathbf{P}\_s\}}
> p(\mathbf{X}^{(a)}\_{i,m}, \mathbf{p} ;\theta)
> $$
>
> Given a $Q(\mathbf{p})$ probability simplex for $\mathbf{p}$,
> such that $\sum\_{\mathbf{p} \in \mathbf{P}} Q(\mathbf{p}) = 1$,
>
> $$
> = \arg\max_\theta
> \sum_{i} \sum\_{a=1}^A \sum\_{m=1}^M
> \log \sum\_{\mathbf{p} \in \{\mathbf{P}\_m \cup \mathbf{P}\_s\}}
> Q(\mathbf{p})
> \frac{ p(\mathbf{X}^{(a)}\_{i,m}, \mathbf{p} ;\theta)}{Q(\mathbf{p})}
> $$
>
> With Jensen's inequality,
>
> $$
> \ge \arg\max_\theta
> \sum\_{i} \sum\_{a=1}^A \sum_{m=1}^M
> \sum\_{\mathbf{p} \in \{\mathbf{P}\_m \cup \mathbf{P}\_s\}}
> Q(\mathbf{p})
> \log \frac{ p(\mathbf{X}^{(a)}\_{i,m}, \mathbf{p} ;\theta)}{Q(\mathbf{p})}
> $$
>
> Therefore, removing constants, we should maximize:
> $$
> \sum\_{i} \sum\_{m=1}^M \sum\_{\textbf{p} \in \{\textbf{P}\_m \cup \textbf{P}\_{s}\}}  Q(\textbf{p}) \log  { p(\textbf{X}^{(a)}\_{i,m} , \textbf{p} ;\theta)}
> $$
>
> $$
> = \sum\_{i} \sum\_{m=1}^M \left( \sum\_{\textbf{p} \in \textbf{P}\_s}  Q(\textbf{p}) \log  { p(\textbf{X}^{(a)}\_{i,m} , \textbf{p} ;\theta)} + \sum\_{\textbf{p} \in \textbf{P}\_m}  Q(\textbf{p}) \log  { p(\textbf{X}^{(a)}\_{i,m} , \textbf{p} ;\theta)} \right)
> $$
> Now with  $\mathbf{U}\_{m}^{(a)}= \mathrm{Softmax}({\mathbf{E}}_{m}^{(a)} \mathbf{P} / \tau)$, the softmax forms a probability distribution over $\mathbf{X}^{(a)}\_{i,m}$ and $\textbf{p}$, so we can substitute it in.
> $$
> = \sum\_{i} \sum\_{m=1}^M \sum\_{a=1}^A \left( \sum\_{\textbf{p} \in \textbf{P}\_s}  Q(\textbf{p}) \log  { \textbf{U}^{(a)}\_m} + \sum\_{\textbf{p} \in \textbf{P}\_m}  Q(\textbf{p}) \log  {\textbf{U}^{(a)}\_m} \right)
> $$
>
> Now with $\mathbf{V}_{m}^{(a)} = \mathrm{Sinkhorn}({\mathbf{E}}\_{m}^{(a)} \mathbf{P})$, the sinkhorn give us normalized soft prototype label assignments, we can substitute them in as being proportional to a true probability simplex.
>
> $$
> \propto \sum\_{i} \sum\_{m,n=1}^M \sum\_{a,b=1}^A \left( \sum\_{n \neq m} \textbf{V}^{(a)}\_n \log \textbf{U}^{(a)}\_m + \sum\_{b \neq a} \textbf{V}^{(b)}\_m \log \textbf{U}^{(a)}\_m \right) = - \mathcal{L}\_{\text{MPP}} * \beta
> $$
>
> This is now equivalent our negative loss function multiplied by a constant $\beta$. Therefore, by minimizing our $\mathcal{L}\_{\text{MPP}}$, we are maximizing this lower bound of the log-likelihood.

---

> ### Author Response · Authors · 2025-11-26
>
> # 1. [Continued] Further investigation into prototypes
> ### Response:
>  [Continued] Without the prototypes, ProtoMM reverts to SLIP, relying on a continuous embedding space optimized via standard contrastive objectives (e.g., SimCLR/CLIP). However, recent literature highlights that such contrastive learning can often fail to learn an overlapping multimodal embedding due to the “modality gap” [11] (a persistent separation in the embedding caused by the contrastive objective encouraging the retention of modality-specific noise) and “class collision" [12, 13] (contrastive learning pushes semantically similar instances because they share a batch and are in the negative set).
>
> ProtoMM unifies these insights to project embeddings onto a finite set of shared prototypes. This acts as a discrete bottleneck that filters out modality-specific semantically-irrelevant noise [14], bridging the modality gap. Simultaneously, it clusters semantically similar instances, resolving class collisions. **This discretization is particularly well-suited for physiological time-series, which are composed of recurring motifs (e.g., neuroactivity patterns [15] or gait cycles [16]) that are naturally encoded by a discrete dictionary.**
>
> # 2. Capturing Within and Between-Modality Information with α
> ### Addressing:
> * W4: “The ablation study is somewhat shallow, as it explores only three α values (0, 0.5, and 1). Testing a finer range would yield more informative results regarding the model’s sensitivity and stability.”
> * Q2: “In equations (5)-(7), there appear to be  individual within-modality losses and  between-modality losses. Would this not still lead to an imbalance between the two loss types, particularly when ?”
> ### Response:
>
> Thank you for the great suggestion. **Originally, we did not want to rely on an extensive hyperparameter tuning in order to achieve strong results, so we chose $\alpha=0.5$ to as a simple way to combine the within and between-modality losses together.** However, we agree that a finer-grained analysis provides valuable intuition into the interplay between the two loss types.
>
> **We have updated our ablation study (Table 2) in the paper [Lines 378-391] to include further ablations on α=[0,0.25,0.5,0.67,0.75,1].** Here we see that ProtoMM achieves the best performance at α=0.5 and 0.67, showing that the unified MPP objective successfully captures both within- and between-modality information. As such, including both loss functions is necessary for learning shared semantic information.
>
> Regarding the imbalance in Equations (5)-(7): You are correct that the raw number of terms in the within-modality losses differs from the between-modality losses. For our setting, this results in 4 within-modality pairs versus 8 between-modality pairs, naturally biasing the loss toward the between-modality objective if left unweighted. As we see, weighing α slightly more to 0.67 will perfectly balance out the two loss-terms and achieve arguably better performance.
>
>
> | Dataset | Task | Metric | α=0 | α=0.25 | α=0.5 (ProtoMM) | α=0.67 | α=0.75 | α=1 |
> | :--- | :--- | :---: | :---: | :---: | :---: | :---: | :---: | :---: |
> | **MOODS** | Stress (2) | F1 (↑) | 0.461±0.005 | 0.463±0.005 | _0.532±0.006_ | 0.522±0.006 | **0.537±0.006** | 0.497±0.006 |
> | | | Acc (↑) | _0.604±0.006_ | **0.608±0.005** | 0.591±0.006 | 0.586±0.006 | 0.600±0.006 | 0.591±0.006 |
> | | Activity (2) | F1 (↑) | 0.719±0.005 | 0.724±0.005 | **0.872±0.003** | 0.869±0.003 | _0.871±0.003_ | 0.855±0.004 |
> | | | Acc (↑) | 0.874±0.002 | 0.876±0.002 | **0.932±0.002** | 0.929±0.002 | _0.931±0.002_ | 0.923±0.002 |
> | **WESAD** | Stress (4) | F1 (↑) | 0.578±0.057 | 0.589±0.058 | **0.622±0.050** | 0.577±0.054 | 0.526±0.047 | _0.612±0.055_ |
> | | | Acc (↑) | 0.663±0.050 | _0.685±0.052_ | **0.719±0.048** | 0.674±0.050 | 0.640±0.051 | _0.685±0.050_ |
> | | Stress (2) | F1 (↑) | 0.783±0.052 | 0.786±0.054 | _0.910±0.035_ | **0.938±0.031** | 0.800±0.051 | 0.858±0.049 |
> | | | Acc (↑) | 0.808±0.045 | 0.822±0.044 | _0.918±0.031_ | **0.945±0.027** | 0.822±0.045 | 0.890±0.037 |
> | **PPG-DaLiA**| Activity (9) | F1 (↑) | 0.496±0.005 | 0.485±0.005 | 0.656±0.004 | _0.657±0.004_ | **0.662±0.004** | 0.618±0.005 |
> | | | Acc (↑) | 0.502±0.004 | 0.493±0.004 | 0.638±0.004 | **0.651±0.004** | _0.648±0.004_ | 0.603±0.004 |
> | | HR (R) | MAE (↓) | 10.45±0.08 | 10.76±0.08 | _8.74±0.07_ | **8.67±0.07** | 9.08±0.08 | 9.14±0.08 |
> | | | R2 (↑) | 0.536±0.009 | 0.515±0.009 | _0.648±0.008_ | **0.654±0.007** | 0.618±0.008 | 0.611±0.008 |
>
> *Note: **Bold** indicates best performance; _Italics_ indicates second best.*

---

> ### Author Response · Authors · 2025-11-26
>
> # 3. Further investigating interpretability
> ### Addressing:
> * W5: “The interpretability evaluation is purely qualitative, lacking quantitative metrics such as clustering purity or correlations with physiological labels. Moreover, interpretability is not compared against baseline models, making it unclear whether these insights are unique to the proposed method.”
> * Q5: “In the interpretability experiment, could the authors provide quantitative validation showing that the learned prototypes correspond to specific physiological states?”
>
> ### Response:
> The reviewer raised a great point about interpretability of the learned prototypes. Going beyond the visualization of the prototypes, we now perform quantitative evaluation of the prototypes by computing their class purity.
>
> For each prototype, we retrieve the k nearest neighbors (from both the PPG and accelerometer modalities), and analyze if these neighbors belong to the same class. This purity score quantifies whether prototypes represent semantic clusters. We show the results of this exploration in Table 10 (Appendix).
> Across datasets, we observe high purity scores especially when k is lower, indicating that prototypes consistently group semantically similar physiological segments. We add the table below for reference.
>
> **Table: Prototype class purity (%) over binary classification tasks.**
> | Dataset | Task | Mod | Top-3 | Top-5 | Top-10 |
> | :--- | :--- | :---: | :---: | :---: | :---: |
> | **MOODS** | Stress (2) | A | 41.95 | 23.92 | 9.21 |
> | | | P | 36.56 | 16.33 | 4.56 |
> | | Activity (2) | A | 83.57 | 77.65 | 70.13 |
> | | | P | 79.80 | 71.16 | 61.15 |
> | **WESAD** | Stress (2) | A | 53.74 | 33.12 | 13.26 |
> | | | P | 64.85 | 45.62 | 27.38 |

---

> ### Author Response · Authors · 2025-11-26
>
> # 4. Clarifying
> ### Addressing:
> * W2 / W3 / W6 / Q1 / Q3 / Q4
>
> ### Response:
>
> * W3: “Several closely related baseline methods (e.g., SLIP, FOCAL) are introduced only in the experimental section, rather than in the related work discussion. Including them earlier would clarify the paper’s positioning and help readers assess the novelty of the proposed approach more fairly.”
>
> Added.
>
> * W6: “Minor typos in lines 058, 073.”
>
> Fixed.
>
> * Q1: “How were the hyperparameters, such as $\alpha$ and the encoder architecture parameters, selected? Were they tuned empirically or determined through heuristic choices?”
>
> Thank you for this discussion point. **In general, we did not want to rely on an extensive hyperparameter tuning in order to achieve strong results, so we chose default parameters while training.** This includes setting sinkhorn_iter=3 and temperature=0.1 to match prior work [1] and $\alpha=0.5$ as a simple way to combine the within and between-modality losses. Additionally, the ResNet1D encoder architecture is set to match prior work as well [2], and it is kept constant across all methodologies to ensure a controlled, fair comparison.
>
> However, we agree that a finer-grained hyperparameter analysis provides valuable information on its robustness, and during the rebuttal, we have run extensive hyperparameter ablation experiments.
>
> **We have updated our ablation study (Table 2) in the paper [Lines 378-391] to include further ablations on α=[0,0.25,0.5,0.67,0.75,1].** Here we see that ProtoMM achieves the best performance at α=0.5 and 0.67, showing that the unified MPP objective successfully captures both within- and between-modality information via a simple balanced combination of the two.
>
> | Dataset | Task | Metric | α=0 | α=0.25 | α=0.5 (ProtoMM) | α=0.67 | α=0.75 | α=1 |
> | :--- | :--- | :---: | :---: | :---: | :---: | :---: | :---: | :---: |
> | **MOODS** | Stress (2) | F1 (↑) | 0.461±0.005 | 0.463±0.005 | _0.532±0.006_ | 0.522±0.006 | **0.537±0.006** | 0.497±0.006 |
> | | | Acc (↑) | _0.604±0.006_ | **0.608±0.005** | 0.591±0.006 | 0.586±0.006 | 0.600±0.006 | 0.591±0.006 |
> | | Activity (2) | F1 (↑) | 0.719±0.005 | 0.724±0.005 | **0.872±0.003** | 0.869±0.003 | _0.871±0.003_ | 0.855±0.004 |
> | | | Acc (↑) | 0.874±0.002 | 0.876±0.002 | **0.932±0.002** | 0.929±0.002 | _0.931±0.002_ | 0.923±0.002 |
> | **WESAD** | Stress (4) | F1 (↑) | 0.578±0.057 | 0.589±0.058 | **0.622±0.050** | 0.577±0.054 | 0.526±0.047 | _0.612±0.055_ |
> | | | Acc (↑) | 0.663±0.050 | _0.685±0.052_ | **0.719±0.048** | 0.674±0.050 | 0.640±0.051 | _0.685±0.050_ |
> | | Stress (2) | F1 (↑) | 0.783±0.052 | 0.786±0.054 | _0.910±0.035_ | **0.938±0.031** | 0.800±0.051 | 0.858±0.049 |
> | | | Acc (↑) | 0.808±0.045 | 0.822±0.044 | _0.918±0.031_ | **0.945±0.027** | 0.822±0.045 | 0.890±0.037 |
> | **PPG-DaLiA**| Activity (9) | F1 (↑) | 0.496±0.005 | 0.485±0.005 | 0.656±0.004 | _0.657±0.004_ | **0.662±0.004** | 0.618±0.005 |
> | | | Acc (↑) | 0.502±0.004 | 0.493±0.004 | 0.638±0.004 | **0.651±0.004** | _0.648±0.004_ | 0.603±0.004 |
> | | HR (R) | MAE (↓) | 10.45±0.08 | 10.76±0.08 | _8.74±0.07_ | **8.67±0.07** | 9.08±0.08 | 9.14±0.08 |
> | | | R2 (↑) | 0.536±0.009 | 0.515±0.009 | _0.648±0.008_ | **0.654±0.007** | 0.618±0.008 | 0.611±0.008 |
>
> *Note: **Bold** indicates best performance; _Italics_ indicates second best.*

---

> ### Author Response · Authors · 2025-11-26
>
> # 4. [Continued] Clarifying
> ### Response:
> * Q1: “How were the hyperparameters, such as $\alpha$ and the encoder architecture parameters, selected? Were they tuned empirically or determined through heuristic choices?”
>
> [Continued] Similarly, we investigate how the number of prototypes influences representation quality (Table 6 in Appendix). **We expand ablation study (Table 6 in Appendix) with number of prototypes P = [0.3k, 1k, 3k, 10k, 30k, 100k]** Across datasets, we observe stable performance when P is greater or equal to 1k. The performance gains for extremely large sets of prototypes (30k, 100k) are small (e.g. 1.1% improvement on MOODS stress F1, 1.2% improvement on WESAD binary stress F1). This confirms that ProtoMM does not rely on large prototype sets to function well. Overall, this demonstrates that ProtoMM is highly robust to prototype-set size.
>
> **Table: Effect of the number of prototypes (P) on downstream performance.**
>
> | Dataset | Task | Metric | P=0.3k | P=1k | P=3k | P=10k (ProtoMM) | P=30k | P=100k |
> | :--- | :--- | :---: | :---: | :---: | :---: | :---: | :---: | :---: |
> | **MOODS** | Stress (2) | F1 (↑) | 0.448±0.005 | _0.532±0.006_ | _0.532±0.006_ | _0.532±0.006_ | 0.529±0.006 | **0.543±0.006** |
> | | | Acc (↑) | **0.616±0.005** | 0.587±0.006 | 0.593±0.006 | 0.591±0.006 | 0.586±0.005 | _0.597±0.006_ |
> | | Activity (2) | F1 (↑) | 0.619±0.005 | _0.871±0.003_ | 0.868±0.003 | **0.872±0.003** | _0.871±0.003_ | 0.869±0.003 |
> | | | Acc (↑) | 0.855±0.002 | _0.931±0.002_ | 0.929±0.002 | **0.932±0.002** | _0.931±0.002_ | 0.930±0.002 |
> | **WESAD** | Stress (4) | F1 (↑) | 0.518±0.059 | **0.641±0.053** | 0.556±0.036 | _0.622±0.050_ | 0.546±0.049 | 0.607±0.053 |
> | | | Acc (↑) | 0.596±0.055 | **0.719±0.047** | _0.697±0.050_ | **0.719±0.048** | 0.618±0.052 | _0.697±0.049_ |
> | | Stress (2) | F1 (↑) | 0.712±0.059 | _0.910±0.035_ | 0.905±0.037 | _0.910±0.035_ | **0.922±0.034** | 0.824±0.048 |
> | | | Acc (↑) | 0.767±0.049 | _0.918±0.032_ | _0.918±0.032_ | _0.918±0.031_ | **0.932±0.030** | 0.836±0.044 |
> | **PPG-DaLiA**| Activity (9) | F1 (↑) | 0.498±0.005 | **0.661±0.004** | _0.656±0.004_ | _0.656±0.004_ | 0.645±0.004 | **0.661±0.004** |
> | | | Acc (↑) | 0.503±0.004 | **0.645±0.004** | 0.634±0.004 | 0.638±0.004 | 0.626±0.004 | _0.641±0.004_ |
> | | HR (R) | MAE (↓) | 10.58±0.08 | **8.59±0.07** | 8.84±0.07 | _8.74±0.07_ | 8.82±0.07 | 8.78±0.07 |
> | | | R2 (↑) | 0.529±0.009 | **0.662±0.007** | 0.641±0.008 | _0.648±0.008_ | 0.638±0.008 | 0.643±0.008 |
>
> *Note: **Bold** indicates best performance; _Italics_ indicates second best.*
>
>
> * Q3: “Could the authors clarify the definitions of $z_t$ and $q_s$ in equation (4)?”
>
> Apologies, this was a typo, equation 4 has now been updated to show
> $$
>    \ell(\mathbf{V}, \mathbf{U}) = - \mathbf{V} \cdot \log \mathbf{U}
> $$
>
> * Q4: “In Lines 361–364, why would the hypothesized advantages of the prototype-based approach not extend to unimodal settings, especially given that empirical results show contrastive methods outperforming the prototype-based variant?”
>
> This is a great discussion point, and we have added this discussion into our Section 5 Results [374-377].
>
> In unimodal settings, SimCLR consistently outperforms its prototype-based counterpart, ProtoMM W-Mod. This aligns with prior work in unimodal time-series self-supervision [3]. We hypothesize this is for two reasons: first, the primary function of the prototypes is to act as a shared, discretized vocabulary that provides a "common language" to translate between disparate data streams (e.g., PPG and Accelerometry). In a unimodal setting, the augmented views already reside on the same manifold, rendering this translation mechanism redundant. Second, the discrete bottleneck acts as a regularizer designed to filter out non-shared, modality-specific noise. In the unimodal case, where such independent noise sources are absent, this discretization acts effectively as a low-pass filter, discarding fine-grained continuous features that SimCLR preserves for richer instance discrimination.
>
>
> [1] Caron, Mathilde, et al. "Unsupervised learning of visual features by contrasting cluster assignments." Advances in neural information processing systems 33 (2020): 9912-9924.
> [2] Saha, Mithun, et al. "Pulse-ppg: An open-source field-trained ppg foundation model for wearable applications across lab and field settings." Proceedings of the ACM on Interactive, Mobile, Wearable and Ubiquitous Technologies 9.3 (2025): 1-35.
> [3] Meng, Qianwen, et al. "MHCCL: masked hierarchical cluster-wise contrastive learning for multivariate time series." Proceedings of the AAAI Conference on Artificial Intelligence. Vol. 37. No. 8. 2023.

---

### Official Review · Reviewer_xWEy · 2025-11-11

**Soundness:** 2
**Presentation:** 3
**Contribution:** 2
**Rating:** 4
**Confidence:** 3

**Summary:**

This paper introduces ProtoMM, a self-supervised learning framework for multimodal time-series
data that addresses the limitations of traditional methods in modality alignment by using a shared
prototype space. ProtoMM outperforms existing methods in several tasks, particularly in stress
detection and activity recognition. Additionally, prototype visualization enhances the model's
interpretability. This approach offers an innovative solution for multimodal self-supervised
learning

**Strengths:**

The ProtoMM framework addresses the issue of negative sample sampling in
multimodal self-supervised learning by introducing a shared prototype space. Particularly in the
application of biosignals, ProtoMM effectively captures complementary information both within
and between modalities, providing a novel solution.


The model framework is highly versatile and can seamlessly be applied to different types
of time-series modalities.

**Weaknesses:**

Although ProtoMM outperforms the existing baseline models on some metrics, its
performance improvement is very limited (about 0.01-0.02), and it is difficult to determine
whether the improvement is due to the effect of the method itself or the experimental
randomness;


 The paper claims that ProtoMM can simultaneously capture within-modality (unique)
and between-modality (shared) information, but does not design a direct and objective
experiment to verify it. The authors only infer indirectly that the model learns on both types
of information based on the optimal model performance when α=0.5, which lacks support.

**Questions:**

How can you prove that ProtoMM can effectively capture intra and inter-modality information?

How do you demonstrate that prototyping is more effective on biological time series data?

---

> ### Author Response · Authors · 2025-11-26
>
> # 1. Significant Performance Increase
> ### Addressing:
> * W1: “Although ProtoMM outperforms the existing baseline models on some metrics, its performance improvement is very limited (about 0.01-0.02), and it is difficult to determine whether the improvement is due to the effect of the method itself or the experimental randomness;”
>
> ### Response:
> We have now performed additional analysis by bootstrapping random subsets of the test set 1000 times and computed the standard deviation of the resulting performance. All tables have now been updated with these values, demonstrating that although performance improvement may seem marginal, the differences are statistically significant.
>
> **Table 1: Results on MOODS Dataset**
>
> | Model | In | Out | Stress (2) F1 (↑) | Stress (2) Acc (↑) | Activity (2) F1 (↑) | Activity (2) Acc (↑) |
> | :--- | :---: | :---: | :---: | :---: | :---: | :---: |
> | **Unimodal** | | | | | | |
> | SimCLR | A | A | 0.477±0.006 | 0.598±0.006 | 0.861±0.004 | 0.925±0.002 |
> | SimCLR | P | P | _0.527±0.006_ | 0.607±0.006 | 0.655±0.005 | 0.857±0.002 |
> | PMM WMod | A | A | 0.464±0.006 | 0.604±0.006 | 0.857±0.004 | 0.924±0.002 |
> | PMM WMod | P | P | 0.469±0.005 | 0.611±0.005 | 0.595±0.005 | 0.848±0.003 |
> | **Multimodal** | | | | | | |
> | SimCLR | P+A | P+A | 0.488±0.006 | 0.601±0.006 | 0.849±0.004 | 0.919±0.002 |
> | PMM WMod | P+A | P+A | 0.467±0.006 | 0.600±0.006 | 0.836±0.004 | 0.913±0.002 |
> | [NEW] Abbaspourazad et al. | P+A | P+A | 0.445±0.005 | _0.614±0.005_ | 0.681±0.005 | 0.865±0.002 |
> | [NEW]  RelCon | P+A | P+A | 0.520±0.006 | 0.589±0.005 | 0.748±0.004 | 0.880±0.002 |
> | CLIP | P\|A | P+A | 0.524±0.006 | 0.578±0.006 | _0.869±0.003_ | _0.929±0.002_ |
> | COCOA | P\|A | P+A | 0.520±0.006 | 0.579±0.005 | 0.858±0.003 | 0.924±0.002 |
> | CroSSL | P\|A | P+A | 0.461±0.005 | **0.622±0.006** | 0.751±0.004 | 0.882±0.002 |
> | FOCAL | P\|A | P+A | 0.510±0.006 | 0.588±0.006 | 0.853±0.003 | 0.921±0.002 |
> | SLIP | P\|A | P+A | 0.524±0.006 | 0.586±0.006 | 0.867±0.003 | _0.929±0.002_ |
> | **ProtoMM** | P\|A | P+A | **0.532±0.006** | 0.591±0.006 | **0.872±0.003** | **0.932±0.002** |
>
> **Table 2: Results on WESAD Dataset**
>
> | Model | In | Out | Stress (4) F1 (↑) | Stress (4) Acc (↑) | Stress (2) F1 (↑) | Stress (2) Acc (↑) |
> | :--- | :---: | :---: | :---: | :---: | :---: | :---: |
> | **Unimodal** | | | | | | |
> | SimCLR | A | A | 0.557±0.053 | 0.629±0.052 | 0.793±0.055 | 0.836±0.044 |
> | SimCLR | P | P | 0.348±0.034 | 0.517±0.053 | 0.692±0.056 | 0.699±0.054 |
> | PMM WMod | A | A | 0.533±0.057 | 0.584±0.053 | 0.675±0.060 | 0.726±0.053 |
> | PMM WMod | P | P | 0.522±0.055 | 0.607±0.053 | 0.847±0.050 | 0.877±0.039 |
> | **Multimodal** | | | | | | |
> | SimCLR | P+A | P+A | 0.445±0.048 | 0.596±0.053 | 0.721±0.056 | 0.753±0.051 |
> | PMM WMod | P+A | P+A | 0.486±0.054 | 0.562±0.055 | 0.753±0.056 | 0.795±0.047 |
> | [NEW]  Abbaspourazad et al.| P+A | P+A | 0.417±0.044 | 0.551±0.056 | 0.668±0.062 | 0.753±0.050 |
> | [NEW]  RelCon | P+A | P+A | 0.573±0.052 | 0.640±0.051 | **0.910±0.035** | **0.918±0.031** |
> | CLIP | P\|A | P+A | 0.496±0.048 | 0.618±0.053 | **0.910±0.034** | **0.918±0.031** |
> | COCOA | P\|A | P+A | 0.508±0.050 | 0.607±0.053 | 0.778±0.053 | 0.808±0.046 |
> | CroSSL | P\|A | P+A | 0.430±0.045 | 0.494±0.052 | 0.783±0.053 | 0.808±0.047 |
> | FOCAL | P\|A | P+A | _0.596±0.049_ | _0.708±0.049_ | 0.888±0.041 | 0.904±0.035 |
> | SLIP | P\|A | P+A | 0.500±0.039 | 0.652±0.051 | 0.848±0.046 | 0.863±0.040 |
> | **ProtoMM** | P\|A | P+A | **0.622±0.050** | **0.719±0.048** | **0.910±0.035** | **0.918±0.031** |
>
> **Table 3: Results on PPG-DaLiA Dataset**
>
> | Model | In | Out | Activity (9) F1 | Activity (9) Acc | HR MAE (↓) | HR R2 (↑) |
> | :--- | :---: | :---: | :---: | :---: | :---: | :---: |
> | **Unimodal** | | | | | | |
> | SimCLR | A | A | 0.559±0.005 | 0.560±0.004 | 15.37±0.10 | 0.101±0.015 |
> | SimCLR | P | P | 0.302±0.004 | 0.399±0.004 | **8.14±0.07** | **0.670±0.008** |
> | PMM WMod | A | A | 0.586±0.005 | 0.577±0.004 | 15.45±0.10 | 0.097±0.015 |
> | PMM WMod | P | P | 0.285±0.004 | 0.397±0.004 | _8.29±0.07_ | **0.670±0.007** |
> | **Multimodal** | | | | | | |
> | SimCLR | P+A | P+A | 0.542±0.005 | 0.536±0.004 | 9.91±0.08 | 0.534±0.009 |
> | PMM WMod | P+A | P+A | 0.543±0.005 | 0.547±0.005 | 15.07±0.09 | 0.155±0.014 |
> | [NEW]  Abbaspourazad et al. | P+A | P+A | 0.444±0.005 | 0.453±0.004 | 15.38±0.10 | 0.103±0.014 |
> | [NEW]  RelCon | P+A | P+A | 0.433±0.005 | 0.461±0.005 | 9.63±0.08 | 0.578±0.009 |
> | CLIP | P\|A | P+A | _0.640±0.005_ | _0.624±0.004_ | 9.37±0.07 | 0.609±0.008 |
> | COCOA | P\|A | P+A | 0.620±0.005 | 0.602±0.004 | 9.43±0.07 | 0.615±0.008 |
> | CroSSL | P\|A | P+A | 0.553±0.005 | 0.540±0.004 | 11.47±0.08 | 0.464±0.010 |
> | FOCAL | P\|A | P+A | 0.607±0.005 | 0.606±0.004 | 11.66±0.08 | 0.449±0.010 |
> | SLIP | P\|A | P+A | 0.633±0.004 | 0.625±0.004 | 8.89±0.08 | 0.634±0.008 |
> | **ProtoMM** | P\|A | P+A | **0.656±0.004** | **0.638±0.004** | 8.74±0.07 | 0.648±0.008 |
>
> *Key: A=Accel, P=PPG; + designates concatenation, \| designates separate encoders.*

---

> ### Author Response · Authors · 2025-11-26
>
> # 2. Capturing Within and Between-Modality Information
> ### Addressing:
> * W2: “The paper claims that ProtoMM can simultaneously capture within-modality (unique) and between-modality (shared) information, but does not design a direct and objective experiment to verify it. The authors only infer indirectly that the model learns on both types of information based on the optimal model performance when α=0.5, which lacks support.”
> * Q1: “How can you prove that ProtoMM can effectively capture intra and inter-modality information?”
>
> ### Response:
> Thank you for bringing up this point. $\alpha$ controls the weights associated with the additive combination of within- and between-modality losses, as seen in equation 7.
> **Originally, we did not want to rely on an extensive hyperparameter tuning in order to achieve strong results, so we chose $\alpha=0.5$ to as a simple way to evenly combine the within and between-modality losses together.** However, we agree that a finer-grained analysis provides valuable intuition into the interplay between the two loss types.
>
> **We have updated our ablation study (Table 2) in the paper [Lines 378-391] to include further ablations on α=[0,0.25,0.5,0.67,0.75,1].** Here we see that ProtoMM achieves the best performance at α=0.5 and 0.67, showing that the unified MPP objective successfully captures both within- and between-modality information. As such, including both loss functions is necessary for learning shared semantic information.
>
> | Dataset | Task | Metric | α=0 | α=0.25 | α=0.5 (ProtoMM) | α=0.67 | α=0.75 | α=1 |
> | :--- | :--- | :---: | :---: | :---: | :---: | :---: | :---: | :---: |
> | **MOODS** | Stress (2) | F1 (↑) | 0.461±0.005 | 0.463±0.005 | _0.532±0.006_ | 0.522±0.006 | **0.537±0.006** | 0.497±0.006 |
> | | | Acc (↑) | _0.604±0.006_ | **0.608±0.005** | 0.591±0.006 | 0.586±0.006 | 0.600±0.006 | 0.591±0.006 |
> | | Activity (2) | F1 (↑) | 0.719±0.005 | 0.724±0.005 | **0.872±0.003** | 0.869±0.003 | _0.871±0.003_ | 0.855±0.004 |
> | | | Acc (↑) | 0.874±0.002 | 0.876±0.002 | **0.932±0.002** | 0.929±0.002 | _0.931±0.002_ | 0.923±0.002 |
> | **WESAD** | Stress (4) | F1 (↑) | 0.578±0.057 | 0.589±0.058 | **0.622±0.050** | 0.577±0.054 | 0.526±0.047 | _0.612±0.055_ |
> | | | Acc (↑) | 0.663±0.050 | _0.685±0.052_ | **0.719±0.048** | 0.674±0.050 | 0.640±0.051 | _0.685±0.050_ |
> | | Stress (2) | F1 (↑) | 0.783±0.052 | 0.786±0.054 | _0.910±0.035_ | **0.938±0.031** | 0.800±0.051 | 0.858±0.049 |
> | | | Acc (↑) | 0.808±0.045 | 0.822±0.044 | _0.918±0.031_ | **0.945±0.027** | 0.822±0.045 | 0.890±0.037 |
> | **PPG-DaLiA**| Activity (9) | F1 (↑) | 0.496±0.005 | 0.485±0.005 | 0.656±0.004 | _0.657±0.004_ | **0.662±0.004** | 0.618±0.005 |
> | | | Acc (↑) | 0.502±0.004 | 0.493±0.004 | 0.638±0.004 | **0.651±0.004** | _0.648±0.004_ | 0.603±0.004 |
> | | HR (R) | MAE (↓) | 10.45±0.08 | 10.76±0.08 | _8.74±0.07_ | **8.67±0.07** | 9.08±0.08 | 9.14±0.08 |
> | | | R2 (↑) | 0.536±0.009 | 0.515±0.009 | _0.648±0.008_ | **0.654±0.007** | 0.618±0.008 | 0.611±0.008 |
>
> *Note: **Bold** indicates best performance; _Italics_ indicates second best.*
>
> # 3. Ties to Biological Time-series Data
> ### Addressing:
> * Q2: “How do you demonstrate that prototyping is more effective on biological time series data?”
>
> ### Response:
> We thank the reviewer for giving us the opportunity to discuss this further. ProtoMM projects its embeddings onto a finite set of shared prototypes. Then, by using them as prediction targets, these prototypes act as semantic cluster centers, encouraging a discretized embedding space, where each embedding seeks to align with specific prototypes.
>
> This discretization is particularly well-suited for physiological time-series, which are composed of recurring motifs that are naturally encoded by a discrete dictionary, lending themselves to a highly expressive embedding space (e.g. neuroactivity patterns [1], gait cycles [2]) . Additionally, discretization can help encourage the model to learn the discrete underlying physiological states that generate biosensors are measuring (e.g. a body in motion would cause increasing PPG and Accelerometry activity, but a body at rest would reflect as lower, stable signals).
>
> # References
> [1] Pradeepkumar, Jathurshan, et al. "Tokenizing Single-Channel EEG with Time-Frequency Motif Learning." NeurIPS 2025 Workshop on Learning from Time Series for Health.
> [2] Haresamudram, Harish, Irfan Essa, and Thomas Ploetz. "Towards learning discrete representations via self-supervision for wearables-based human activity recognition." Sensors 24.4 (2024): 1238.

---

### Author Response · Authors · 2025-12-03

Dear Area Chair,

We’d like to thank all of the reviewers for taking the time and effort to provide such high quality feedback. We published our responses before the discussion freeze, but unfortunately, none of our reviewers were able to respond in time.

Throughout the rebuttal, **we have added 9 new experiments, new mathematical and theoretical derivations, 2 new visualizations, and revised our manuscript’s text for enhanced clarity. We strongly believe that our reviewers would have raised their score given a full discussion period.** Our rebuttal is summarized below.

### 1. Significant Performance Gains & Updated Baselines (Reviewers xWEy, hWpB, 1JH9)
* **Statistical Significance:** To address concerns regarding marginal improvements (xWEy, hWpB), we performed bootstrapping (1,000 runs) on the test set. Table 1 now contains the standard deviations, confirming that ProtoMM’s performance gains are statistically significant across datasets.
* **Modern Baselines:** Responding to Reviewer 1JH9, we added unimodal SSL methods, RelCon [1] and Abbaspourazad et al. [2], as baselines (adapted via early fusion for fairness). ProtoMM consistently outperforms these methods (e.g., achieving the highest F1 scores on MOODS, WESAD, and PPG-DaLiA), reinforcing our state-of-the-art status.
* **Generalization to New Modalities:** We validated ProtoMM on a completely new dataset, Moving Object Detection (MOD) [5], which contains seismic vibration and audio (Table 5). ProtoMM outperforms all baselines (CLIP, COCOA, CroSSL, FOCAL, SLIP) in this new setting, proving its generalization capability beyond biosignals.

### 2. Theoretical Grounding of Prototypes (Reviewers rgHi, 1JH9)
* **Mathematical Derivation:** We added a formal proof in Appendix Section A.4, deriving our Multimodal Prototype Prediction loss from the log-likelihood objective. We show that our objective formally maximizes the ELBO of the joint data distribution while the prototypes act as an information bottleneck to filter out semantically-irrelevant noise.
* **Why Prototypes Work (vs. Contrastive Loss):** We clarified that without prototypes, ProtoMM reverts to SLIP (continuous contrastive learning), which suffers from the "modality gap" [6] and "class collision" [7,8]. ProtoMM projects embeddings onto a finite set of shared prototypes, creating a discrete bottleneck. This effectively filters out modality-specific semantically-irrelevant noise [9] and bridges the modality gap.
* **Relevance to Biosignals:** As noted in our response to xWEy, this discretization is particularly well-suited for physiological time-series (e.g., neuroactivity patterns [10], gait cycles [11]), which consist of recurring motifs naturally encoded by a discrete dictionary.

### 3. Deep Dive into Prototypes: Visualization, Stability, and Purity (Reviewer 1JH9, rgHi)
* **Disentanglement (Figure 4):** We added Figure 4, demonstrating that our shared prototype space disentangles information: most prototypes focus on modality-agnostic information, while some will capture modality-specific features. This is beneficial because the prototypes mostly capture the shared underlying generative state, but still retains information that may be more relevant to a given modality (i.e. quasiperiodicity is relevant for PPG but not Accel).
* **Stability (Figure 5):** We addressed concerns about initialization sensitivity by visualizing the prototype space across 10 random seeds (0–900). The scatter patterns remain nearly identical, confirming that the learned prototypes form a stable structure independent of initialization.
* **Interpretability (Table 9):** We computed Class Purity scores for the prototypes. The high purity scores quantitatively confirm that prototypes consistently group semantically similar physiological segments (e.g., distinct activity classes).
* **Sinkhorn Importance (Table 7):** We demonstrated that the Sinkhorn-Knopp normalization is critical. Removing it or replacing it with Softmax leads to significant performance drops (e.g., 30.1% and 26.9% F1 drop on PPG-DaLiA Activity), proving its role in preventing collapse.

---

> ### Author Response · Authors · 2025-12-03
>
> [Continued]
>
> ### 4. Hyperparameters & Experimental Design (Reviewers xWEy, hWpB, rgHi, 1JH9)
> * **Hyperparameter Ablation Experiments** We conducted a fine-grained ablation of $\alpha$ (Table 2). Results show that a balanced combination ($\alpha=0.5$ or $0.67$) of within-modality and between-modality losses is essential for optimal performance. We expanded ablation study with the number of prototypes (Table 6 in Appendix), confirming that ProtoMM is highly robust to prototype-set size.
> * **Computational Overhead:** We clarified that ProtoMM is efficient. The prototype objective scales linearly as $O(EBP)$, whereas standard CLIP-style contrastive losses scale quadratically as $O(EB^2)$. Our ablation (Table 6) shows ProtoMM is effective with moderate prototype sizes (1k–10k).
> * **Fine-tuning vs. Linear Probing:** We performed full fine-tuning experiments (Table 4). The results show that fine-tuning leads to overfitting and performance degradation compared to linear probing. This justifies our use of linear probing (standard in foundation model literature [1,2,3,4]) to isolate the quality of the robust, pre-trained representation.
>
> We believe these revisions comprehensively address the reviewers' concerns and firmly establish the validity of ProtoMM.
>
>
> Best regards,
>
> The Authors
>
>
>
>
> ### References
> [1] Xu et. al., RelCon: Relative Contrastive Learning for a Motion Foundation Model for Wearable Data, ICLR 2025
> [2] Abbaspourazad, et. al., Large-scale Training of Foundation Models for Wearable Biosignals, ICLR 2024
> [3] Narayanswamy et. al., Scaling wearable foundation models, ICLR 2025
> [4] Saha et. al., Pulse-PPG: An Open-Source Field-Trained PPG Foundation Model for Wearable Applications across Lab and Field Settings, ACM on Interactive, Mobile, Wearable and Ubiquitous Technologies
> [5] Liu, Shengzhong, et al. "Focal: Contrastive learning for multimodal time-series sensing signals in factorized orthogonal latent space." Advances in Neural Information Processing Systems 36 (2023): 47309-47338.
> [6] Liang, Victor Weixin, et al. "Mind the gap: Understanding the modality gap in multi-modal contrastive representation learning." Advances in Neural Information Processing Systems 35 (2022): 17612-17625.
> [7] Li, J., et al. "Prototypical Contrastive Learning of Unsupervised Representations." ICLR, 2021.
> [8] Wang, Yifei, et al. "Chaos is a ladder: A new theoretical understanding of contrastive learning via augmentation overlap." ICLR (2022).
> [9] Xia, Yan, et al. "Achieving cross modal generalization with multimodal unified representation." Advances in Neural Information Processing Systems 36 (2023): 63529-63541.
> [10] Pradeepkumar, Jathurshan, et al. "Tokenizing Single-Channel EEG with Time-Frequency Motif Learning." NeurIPS 2025 Workshop on Learning from Time Series for Health.
> [11] Haresamudram, Harish, Irfan Essa, and Thomas Ploetz. "Towards learning discrete representations via self-supervision for wearables-based human activity recognition." Sensors 24.4 (2024): 1238.

---

### Meta-Review · Area_Chair_FH6j · 2026-01-07

**Summary:**

This submission was reviewed by four expert reviewers, with the ratings of: 1 reject, 2 borderline reject, and 1 borderline accept.The major concerns from the reviewers are about the marginal performance gain over baselines, missing convincing support/validation for the claims (overclaims), limited theoretical grounding and missing details of the proposed techniques, unclear motivation, insufficient experimental analysis, interpretability, outdated baselines, datasets issue, the frozen encoders setting, and failure cases. The authors provided a rebuttal for the concerns, but no further response from the reviewer was presented, and there was no discussion.

After carefully going through all the review comments and the authors' rebuttal, it can be seen that some of the concerns and questions are addressed by the authors' further response and experiments. However, there are still outstanding major concerns that are not well addressed. There is no strong support for a clear accept. As a result, it is unfortunate that the paper in its current form is not ready for publication at ICLR, and needs a major revision followed by another round of review for assessment.

**Reviewer Concerns:**

Concerns that the AC thinks were addressed by the rebuttal: missing discussion to related works; shallow ablation study on the $\alpha$ values; quantitative evaluation of interpretability; the outdated baselines concern was partially addressed; fine-tuning experiment was provided (but the concern from the reviewer was not clearly addressed); failure cases; and other minor issues.

Concerns that are still outstanding: the marginal performance gain; claims without convincing validation; lack of convincing theoretical justification; unclear motivation of the proposed method; outdated baselines; modality overclaiming; the datasets issue; the frozen-encoders setting justification; visualizations to support a better understanding.

**Reviewer Scores:**

According to the review comments, and the rebuttal, for each review the reviewer might have changed their score in the way below, if they had been able to participate fully in the discussion:
* Reviewer xWEy: borderline reject to reject
* Reviewer rgHi: borderline accept to borderline reject
* Reviewer hWpB: borderline reject, unchanged
* Reviewer 1JH9: reject, unchanged.

---

### Decision · Program_Chairs · 2026-01-26

Reject